# ON HARMONIZING IMPLICIT SUBPOPULATIONS

**Feng Hong**[1]  **Jiangchao Yao**[1,2,✉]  **Yueming Lyu**[3,4]
**Zhihan Zhou**[1]  **Ivor W. Tsang**[3,4,5]  **Ya Zhang**[1,2]  **Yanfeng Wang**[1,2,✉]

[1]Cooperative Medianet Innovation Center, Shanghai Jiao Tong University

[2]Shanghai Artificial Intelligence Laboratory   [3]CFAR, Agency for Science, Technology and Research

[4]IHPC, Agency for Science, Technology and Research   [5]Nanyang Technological University

{feng.hong, Sunarker, zhihanzhou, ya_zhang, wangyanfeng}@sjtu.edu.cn
{Lyu_Yueming, ivor_tsang}@cfar.a-star.edu.sg

## ABSTRACT

Machine learning algorithms learned from data with skewed distributions usually suffer from poor generalization, especially when minority classes matter as much as, or even more than majority ones. This is more challenging on class-balanced data that has some hidden imbalanced subpopulations, since prevalent techniques mainly conduct class-level calibration and cannot perform subpopulation-level adjustments without subpopulation annotations. Regarding implicit subpopulation imbalance, we reveal that the key to alleviating the detrimental effect lies in effective subpopulation discovery with proper rebalancing. We then propose a novel subpopulation-imbalanced learning method called Scatter and HarmonizE (SHE). Our method is built upon the guiding principle of *optimal data partition*, which involves assigning data to subpopulations in a manner that maximizes the predictive information from inputs to labels. With theoretical guarantees and empirical evidences, SHE succeeds in identifying the hidden subpopulations and encourages subpopulation-balanced predictions. Extensive experiments on various benchmark datasets show the effectiveness of SHE. The code is available.

## 1   INTRODUCTION

The imbalance nature inherent in real-world data challenges algorithmic robustness especially when minority classes matter as much as, or even more than majority ones (Reed, 2001; Zhang et al., 2023b). It becomes more exacerbated in scenarios where the observed categories are apparently balanced but the implicit subpopulations[1] remain imbalanced (Zhang et al., 2020). Specifically, such imbalance stays not in the class level but in the implicit subpopulation level, giving rise to the subpopulation imbalance problem. It is ubiquitous in some sensitive applications, *e.g.*, medical diagnosis with ethnic minorities or auto-driving decisions in rare weathers, yielding severe fairness concerns and generalization impairments (Yang et al., 2023).

Typical studies in imbalanced learning (Buda et al., 2018; He & Garcia, 2009; Wang et al., 2021; Menon et al., 2021; Cui et al., 2021) focus on the class-imbalance setting like Fig. 1(a), employing the explicit class distribution to calibrate the training of majority and minority classes, which cannot handle implicit subpopulation imbalance like Fig. 1(b). Other efforts for spurious correlations, which arise from discrepancies in class distribution across specific attributes compared to the overall class distribution, aim to make predictions by causally relevant features, while excluding these spuriously correlated attributes  (Nam et al., 2020; Zhang et al., 2022; Seo et al., 2022; Taghanaki et al., 2022). Our goal for implicit subpopulation imbalance, shares the similar rebalancing spirit with these works for class imbalance and spurious correlations, but differs in the underlying problems and mechanisms. We present a comprehensive comparison of these three concepts of imbalanced learning in Tab. 1.

The key challenges to cope implicit subpopulation imbalance problems are twofold. First, the mixed distribution of multiple subpopulations makes predictions more difficult (compared to a single

---

[1]In this paper, the term "subpopulations" pertains to some implicit attributes that differentiate the "classes" concept and contribute to intra-class variations.

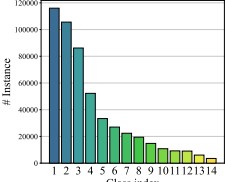 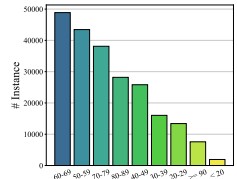 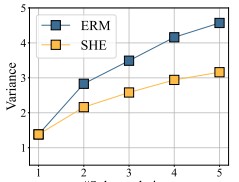 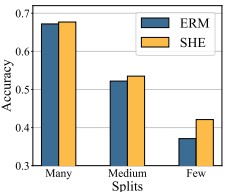

(a) Class imbalance    (b) Subpopulation imbalance    (c) Feature variance    (d) Per-split accuracy

Figure 1: (a) The number of samples for each category in CheXpert (Irvin et al., 2019). The class index is sorted by sample numbers in descending order. The imbalance phenomenon of classes is evident. (b) The imbalanced age subpopulation distribution in CheXpert (Irvin et al., 2019) with the prediction target of diseases. (c) Within-class feature variance at different subpopulation numbers. All experiments are conducted on CIFAR-100 with an imbalance ratio IR = 100, and the within-class variance is calculated as in Papyan et al. (2020). As a comparison, the within-class variance of our method for the learned subpopulations is much lower than ERM under the mixed distribution. (d) Many/Medium/Few accuracies of ERM and SHE in COCO. The performance of minority subpopulations is poor, and our method relatively alleviates this phenomenon.

distribution). This is because vanilla classification models tend to map all training samples of the same class to identical features (Papyan et al., 2020; Han et al., 2022). However, when there are significant discrepancies within a class (*i.e.*, sampling from different subpopulations), forcing them to identical features encounters more obstacles (as illustrated in Fig. 1(c)) and would impair generalization (Ma et al., 2023a). Second, different subpopulations might have different prediction mechanisms (*i.e.*, rely on different features) and machine learning algorithms tend to ignore minority subpopulations, resulting in degraded performance on these data (as in Fig. 1(d)). Besides, the implicit nature of subpopulations makes it harder to conduct rebalancing among subpopulations. These difficulties restrict existing methods from achieving practical effectiveness directly.

To address the above challenges, we propose a novel method to handle implicit subpopulation imbalance, namely, Scatter and HarmonizE (SHE). Intuitively, we seek to decompose complex mixed training data into multiple simpler subpopulations, where the prediction mechanisms within each subpopulation are consistent (Scatter), and then conduct subpopulation balancing (Harmonize). Specifically, we first introduce the concept of *optimal data partition*, which divides training data into subpopulations that can bring the maximum additional prediction ability (Def. 3.1). Then, an empirical risk that is theoretically consistent with the pursuit of optimal data partition (Eq. (1) and Thm. 3.3), is proposed. To account for the imbalance nature of subpopulations, we obtain subpopulation-balanced predictions *w.r.t.* the learned data partition by simply applying the LogSumExp operation to outputs (Thm. 3.4). Finally, a practical realization that can be optimized end-to-end without increasing model capacity is provided (Sec. 3.4). We summarize the contributions as follows:

- We study the practical yet under-explored subpopulation imbalance learning problem that cannot be efficiently solved by existing methods, and identify the unique challenges, whose key lies in exploring the implicit subpopulations to facilitate prediction and subpopulation balancing.
- We proposed a novel SHE method that uncovers hidden subpopulations by optimizing the prediction ability and achieves subpopulation-balanced predictions by simply applying a LogSumExp operation. Theoretical analysis shows promise of SHE under implicit subpopulation imbalance.
- We conduct extensive experiments to comprehensively understand the characteristics of our proposed SHE, and verify its superiority in improving subpopulation imbalance robustness.

## 2 RELATED WORK

In this section, we briefly review the related works developed for the typical class imbalance and spurious correlations, which we summarize as a comparison with our work in Tab. 1.

**Class Imbalance.** Re-sampling (Buda et al., 2018; Wallace et al., 2011) and Re-weighting (Menon et al., 2013; He & Garcia, 2009) are the most widely used methods to train on class-imbalanced datasets. Explorations inspired by transfer learning (Chu et al., 2020; Wang et al., 2021) seek to transfer knowledge from head classes to tail classes to obtain a more balanced performance. Menon et al. (2021); Ren et al. (2020) propose logit adjustment (LA) techniques that modify the output logits by the class-conditional offset terms. The vector-scaling (VS) loss (Kini et al., 2021) instead of considering the simple additive operation, uses multiplicative factors to adjust the output logits. Ma et al. (2023b) proposes to use the semantic scale measured by the feature volume rather than

Table 1: A comparison of different types of imbalance problems, including class-level shifts, subpopulation-level shifts, assumptions underlying the problem and possible negative impacts. For class imbalance, the training class distribution is skewed, *i.e.*, $p_Y(y) \gg p_Y(y')$, where $y = \arg\max_{y \in \mathcal{Y}} p_Y(y), y' = \arg\min_{y \in \mathcal{Y}} p_Y(y)$. For spurious correlation, it is assumed that subpopulations and classes are causally independent but there exists $s \in \mathcal{S}$ that is spuriously correlated with class $y \in \mathcal{Y}$ in training. For subpopulation imbalance, the subpopulation distribution of training data is imbalanced, *i.e.*, $p_S(s) \gg p_S(s')$, where $s = \arg\max_{s \in \mathcal{S}} p_S(s), s' = \arg\min_{s \in \mathcal{S}} p_S(s)$. For simplicity, we use $p(\cdot)$ without subscripts in the following sections to adapt to various variables.

| Imbalance type | Subpopulation shift | Class shift | Assumption | Detrimental Impact on prediction |
|---|---|---|---|---|
| Class Imbalance | – | $p_Y(y) \gg p_Y(y')$ | – | Predict minority classes as majority classes |
| Spurious Correlation | $p_{Y\|S}(y\|s) \gg p_Y(y)$ | – | $S \perp\!\!\!\perp Y$ | Predict relying on irrelevant features |
| Subpopulation Imbalance | $p_S(s) \gg p_S(s')$ | – | – | Ignore features for minority subpopulations |

the sample size of classes to guide the class rebalancing. Cui et al. (2021); Zhu et al. (2022) further improve the prediction performance under class imbalanced data by combining the contrastive learning techniques. Some work (Zhou et al., 2022; 2023; Hong et al., 2023; Zheng et al., 2024) has explored overcoming class imbalance in the context of unsupervised or weakly supervised learning.

**Spurious Correlations.** The distributionally robust optimization (DRO) framework (Ben-Tal et al., 2013; Gao et al., 2017; Duchi et al., 2021) has been proposed to improve the worst case generalization. However, the DRO objective results in excessive attention to worst cases, even if they are implausible. Group DRO (GDRO) (Sagawa et al., 2019) optimizes a soft version of worst-case performance over a set of subgroups, which despite effectiveness requires prior subgroup labels available. Some efforts (Nam et al., 2020; Zhang et al., 2022; Seo et al., 2022) have been made to reduce the reliance on the group-level supervision, but primarily focus on mitigating *spurious correlation* instead of the imbalance among causal factors, namely, removing the false associations between labels and *irrelevant* features in training samples. The typical scheme is first detecting a minority group and then designing an algorithm to promote the detected minority group. Following this framework, a series of works (Nam et al., 2020; Liu et al., 2021; Zhang et al., 2022) explore the minority discovery, which assumes that ERM models are prone to rely on spuriously correlated attributes for prediction, and therefore the failure samples are the minority ones. Some other works (Sohoni et al., 2020; Seo et al., 2022; Liu et al., 2023) treat the model predictions or feature clustering results directly as spuriously correlated features, which in combination with ground-truth can yield more fine-grained subgroup labels. MaskTune (Taghanaki et al., 2022) forces the trained model for more feature exploration by masking, to indirectly mitigate spurious correlations.

## 3 METHOD

### 3.1 PROBLEM FORMULATION

Let $\mathcal{X}$ be the input space and $\mathcal{Y} = \{1, 2, ..., C\}$ be the class space. We denote the underlying space of subpopulations as $\mathcal{S} = \{1, 2, ..., K\}$. The overall data distribution can be formulated as a mixture of distributions of latent subpopulations, *i.e.*, $p(\boldsymbol{x}, y) = \sum_{s \in \mathcal{S}} p(s) \cdot p(\boldsymbol{x}, y|s)$. The training set can be denoted as $\mathcal{D} = \{(\boldsymbol{x}_i, y_i, s_i)\}_{i=1}^N \in (\mathcal{X}, \mathcal{Y}, \mathcal{S})^N$, where any input $\boldsymbol{x}_i$ is associated with a classification label $y_i$ and an *unobserved* subpopulation label $s_i$. Here we focus on the implicit subpopulation imbalance problem, *i.e.*, $p(s)$ is skewed. We assume that subpopulations are heterogeneous with inconsistant predictive mechanisms. That is, data distribution $p(\boldsymbol{x}, y|s)$ differs across subpopulations, and $p(y|\boldsymbol{x}, s)$ may vary among certain subpopulations. For fair evaluation among all subpopulations, a *subpopulation-balanced* test distribution $p_{bal}(\boldsymbol{x}, y) = \sum_{s \in \mathcal{S}} p_{bal}(s) p(\boldsymbol{x}, y|s)$, where $p_{bal}(s) = \frac{1}{K}, \forall s \in \mathcal{S}$, is used for evaluation following imbalanced learning literatures (Menon et al., 2021; Cao et al., 2019). In a nutshell, the goal is to learn a deep model $f : \mathcal{X} \to p(\mathcal{Y})$ on $\mathcal{D}$ that minimizes the following subpopulation-balanced error rate (SBER):

$$\min_f \text{SBER}(f) = \mathbb{E}_{(\boldsymbol{x}, y) \sim p_{bal}(\boldsymbol{x}, y)} \mathbf{1}(y \neq \arg\max_{y' \in \mathcal{Y}} f^{y'}(\boldsymbol{x})).$$

In our experiments, we use a subpopulation-balanced test set as an unbiased estimator for SBER.

### 3.2 MOTIVATION

In Fig. 2, we visualize a toy motivating example whose prediction goal is to distinguish between circles (semi-transparent) and triangles (non-transparent). For training data, they are sampled from

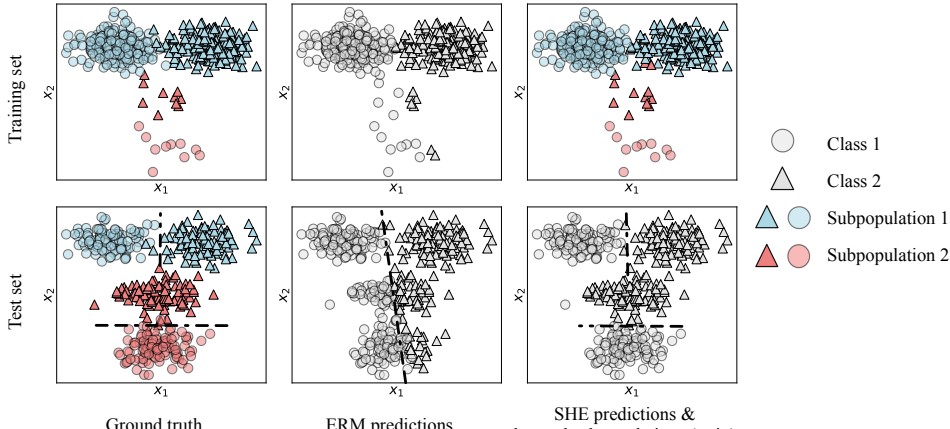

Figure 2: Visualization of a toy motivating example, which is a 2D subpopulation-imbalanced learning problem. The left column illustrates the data distribution of the training set and that of the test set under 2 classes consisting of 2 subpopulations. The middle column exhibits the model prediction of ERM. The right column shows the predictions and the learned subpopulations of SHE on the training set and predictions on the test set. The training set is highly subpopulation-imbalanced with the imbalance ratio IR = 20 and the test set is balanced (referring to Appx. F.1 for more details).

both Subpopulation 1 (blue) and Subpopulation 2 (red), and the training samples of Subpopulation 2 are much less than those of Subpopulation 1, *i.e.,*, under subpopulation imbalance. About the test set, it is balanced sampled from both subpopulations, *i.e.,*, under subpopulation balance[2]. According to the visualization in Fig. 2, $x_1$ is a more important feature in the class prediction for Subpopulation 1, while in terms of Subpopulation 2, $x_2$ can be a more effective feature in the class prediction. Unfortunately, due to the subpopulation imbalance, ERM's predictions rely heavily on $x_1$ and perform poorly in Subpopulation 2. However, if we can accurately identify the latent subpopulations in the training data, such a classification problem in a mixed distribution can be transformed into two simple linear classification problems, and the key features in Subpopulation 2 will not be ignored. Therefore, the key to alleviating subpopulation imbalance is to discover the potential subpopulations in the training data that promote prediction and subpopulation rebalancing. In the right column of Fig. 2, we present the predictions and the learned subpopulations of SHE on the training set and the corresponding predictions on the test set. As can be seen, SHE successfully discriminates between two subpopulations on the training data, with the aid of which more accurate predictions are obtained.

### 3.3 SCATTER AND HARMONIZE

**Optimal Data Partition.** For data with implicit heterogeneous structures, we resort to a proper data partition so that each partition has a consistent predictive mechanism during training. Such a way promotes the prediction ability and helps protect vulnerable subpopulations. In the following, we first introduce the optimal data partition in Def. 3.1 that learns to assign samples to subpopulations.

**Definition 3.1** ((Optimal) Data Partition). Let $X$ and $Y$ be random variables that take values in $\mathcal{X} \times \mathcal{Y}$ following a fixed joint distribution $p_{X,Y}$. A data partition is defined as a mapping $\nu$ of the training data and its labels to the subpopulation space, *i.e.*, $\nu : \mathcal{X} \times \mathcal{Y} \to \mathcal{S}$. So $\nu(X, Y)$ is a random variable taking values from $\mathcal{S}$ and $|\mathcal{S}| = K$. We then define the optimal data partition based on information theory as

$$\nu^* = \arg\max_\nu I(X; Y; \nu(X, Y)) = \arg\max_\nu I(X; Y | \nu(X, Y)) - I(X; Y),$$

where $I(X; Y; \nu(X, Y))$ denotes the interaction information (McGill, 1954) of $X, Y, \nu(X, Y)$, $I(X; Y)$ denotes the mutual information of $X$ and $Y$, and $I(X; Y | \nu(X, Y))$ denotes the conditional mutual information between $X$ and $Y$ given $\nu$.

In information theory, the mutual information $I(X; Y)$ can characterize the prediction ability from input $X$ to class label $Y$ (Cover & Thomas, 2006). The interaction information $I(X; Y; \nu(X, Y))$ means the gain of correlation between $X$ and $Y$ given a data partition $\nu$. A larger $I(X; Y; \nu(X, Y))$ indicates a greater improvement in the prediction ability of a data partition $\nu$ from input $X$ to label

---

[2]In practice, it is common to have a subpopulation-imbalanced set for training. And for the test set, we need to build a subpopulation-balanced counterpart to evaluate the algorithmic robustness *w.r.t.* latent subpopulations.

$Y$. Due to the hierarchical nature of semantics (Deng et al., 2009), the data partition usually comes with multiple possibilities. Def. 3.1 helps us pursue the optimal data partition $\nu^*$ to maximize the prediction ability of the training data. Intuitively, the optimal data partition decomposes the prediction in a complex mixed distribution into several classification problems in multiple simple distributions partitioned by $\nu^*$. In the following, we remark an advantageous property of the optimal data partition.

**Proposition 3.2.** *The optimal data partition at least does not inhibit the prediction ability,* i.e., $I(X;Y;\nu^*(X,Y)) \geq 0$.

Prop. 3.2 shows that the optimal data partition can help to improve the prediction ability, and at least has no negative impact even in the worst case. Please refer to Appx. C.1 for the proof.

**Objective.** After introducing the above concept and analysis, we explore incorporating the idea of optimal data partition to improve the prediction ability and achieve a subpopulation-balanced model. For this reason, we propose the following empirical risk with respect to the training set $\mathcal{D}$, whose relation with the optimal data partition will be proved and discussed in the subsequent theorem.

$$\hat{\mathcal{R}}(f,\nu;\mathcal{D}) = -\frac{1}{N}\sum_{i=1}^{N}\sum_{s\in\mathcal{S}}\mathbf{1}(\nu(\boldsymbol{x}_i,y_i)=s)\cdot\log f_s^{y_i}(\boldsymbol{x}_i) - \hat{H}_{\mathcal{D}}(Y|\nu(X,Y)), \tag{1}$$

where $\mathbf{1}(\cdot)$ denotes the indicator function, with value 1 when $\cdot$ is true and 0 otherwise, $f_s(\boldsymbol{x})$ is the prediction of $\boldsymbol{x}$ for subpopulation $s$, i.e., $f_s : \mathcal{X} \to p(\mathcal{Y})$, and $\hat{H}_{\mathcal{D}}(Y|\nu(X,Y))$ is the empirical entropy of labels conditioning on the data partition $\nu$ with respect to the training set $\mathcal{D}$. We use the following Thm. 3.3 to discuss the consistency between Eq. (1) and the optimal data partition.

**Theorem 3.3.** *Let $f^\dagger = \arg\min_f \hat{\mathcal{R}}(f,\nu;\mathcal{D})$ be the optimal solution for the empirical risk $\hat{\mathcal{R}}(\mathcal{D})$ in Eq. (1) for any $\mathcal{D}$ and $\nu$. Assume that the hypothesis space $\mathcal{H}$ satisfies $\forall \boldsymbol{x} \in \mathcal{X}, \forall y \in \mathcal{Y}, \forall f \in \mathcal{H}, \log f^y(\boldsymbol{x}) > -m$, where $m > 0$. Define a mapping family $G = \{g : \mathcal{X} \times \mathcal{Y} \to \mathbb{R} | g(\boldsymbol{x},y) = \log f^y(\boldsymbol{x}), f \in \mathcal{H}\}$ and $R_N(G) = \mathcal{O}(\frac{1}{\sqrt{N}})$ denotes the Rademacher complexity of $G$ with the sample size $N$ (Bartlett & Mendelson, 2002) (detailed in Appx. B.3). Then for any $\delta \in (0,1)$, we have:*

$$|(I(X;Y;\nu(X,Y))) - (-\hat{\mathcal{R}}(f^\dagger,\nu;\mathcal{D}) + B)| \leq \frac{m}{\sqrt{N}}\sqrt{-2\log\delta} + 4K\cdot R_N(G),$$

*with probability at least $1 - \delta$, where $B = -I(X;Y)$ is a constant, and $K$ is the number of subpopulations.*

Thm. 3.3 presents an important implication that minimizing the empirical risk $\hat{\mathcal{R}}$ in Eq. (1) asympotitically aligns with the direction of maximizing $I(X;Y;\nu(X,Y))$ in Def. 3.1 in a sense of statistical consistency. We kindly refer the readers to Appx. C.2 for the complete proof. To further verify this, we trace the Normalized Mutual Information (NMI) score (Strehl & Ghosh, 2002) between the learned subpopulations and the true subpopulation annotations during training in each epoch and visualize it in Fig. 3. It can be seen that our method gradually learns the sub-populations that correlates well to the true annotations.

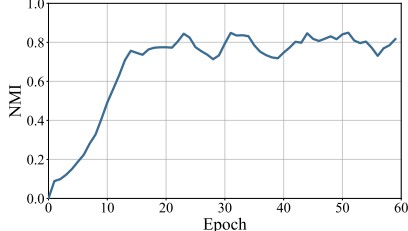

Figure 3: NMI scores between the learned subpopulations and the true annotations on the toy dataset in Fig. 2 during training.

We also visualize the two subpopulations learned by our method in COCO in Fig. 5 in Appendix. It can be observed that our method uncovers meaningful subpopulations, i.e., Subpopulation 1: cut up apples or bananas; Subpopulation 2: the whole apples or bananas. Fig. 3 and Fig. 5 demonstrate the promise of SHE to discover the latent subpopulation structure inherent in the training samples.

**Subpopulation-balanced prediction.** With the inferred subpopulations, we discuss how to achieve subpopulation-balanced predictions. Let $z_s(\boldsymbol{x})$ be the output logits of $\boldsymbol{x}$ for any subpopulation $s \in \mathcal{S}$ and $f_s(\boldsymbol{x}) = \text{softmax}(z_s(\boldsymbol{x}))$. We show that the overall prediction $f(\boldsymbol{x}) = \text{softmax}(z(\boldsymbol{x}))$ with $z(\boldsymbol{x}) = \log\sum_{s\in\mathcal{S}}e^{z_s(\boldsymbol{x})}$ is subpopulation-balanced according to the following Thm. 3.4.

**Theorem 3.4.** *Supposing that for any subpopulation $s \in \mathcal{S}$, $z_s$ can perfectly fit the data distribution of a given subpopulation $s$, i.e., $p(\boldsymbol{x},y|s) \propto e^{z_s^y(\boldsymbol{x})}$, then $z = \log\sum_{s\in\mathcal{S}}e^{z_s}$ can perfectly fit the subpopulation-balanced overall distribution, i.e., $p_{bal}(\boldsymbol{x},y) \propto e^{z^y(\boldsymbol{x})}$.*

Thm. 3.4 implies that alongside pursuing the optimal data partition, the LogSumExp operation on the logits of the learned subpopulations can be directly aggregated into a balanced prediction. We

kindly refer readers to Appx. C.3 for more details. By contrast, the ordinary learning methods will fit the distribution $p(\boldsymbol{x}, y) = \sum_{s \in \mathcal{S}} p(s) \cdot p(\boldsymbol{x}, y|s)$, which is non-robust to subpopulation imbalance.

**Discussion.** We would like to briefly discuss the core differences between SHE and some related techniques. Classic clustering methods (Cheng, 1995; Asano et al., 2020; Caron et al., 2020) divide the input space $\mathcal{X}$ into several disjoint clusters, with the goal that the clusters match as closely to the target classes. Our method, on the other hand, divides the data in a subpopulation level instead of the class level, with the goal that the partition maximally intervenes with predictions from input to classes. Some works for spurious correlations (Sohoni et al., 2020; Seo et al., 2022; Liu et al., 2023) use the predictions of ERM or their feature clustering results as subpopulations, based on an underlying assumption that data from the same subpopulation will have the same ERM predictions or features and conversely not. Such an assumption might not be valid, especially when there are not many spurious associations captured during training. In this case, the clustering learned by these methods remains at the class level, as the ERM model uses the given classes as supervision. In comparison, SHE has theoretically and empirically been oriented to learn meaningful subpopulation structures.

### 3.4 REALIZATION

**Optimization for the data partition $\nu$.** We use a subpopulation-weight matrix $V \in \{V | V \in \mathbb{R}_+^{N \times K}, \text{s. t.} \sum_{s=1}^{K} v_{is} = 1, \forall i = 1, 2, \ldots, N\}$ to represent a data partition $\nu$ in Eq. (1) with respect to the training set $\mathcal{D}$. Each $v_{is}$ in $V$ denotes the probability of the $i$-th data point being sampled from the subpopulation $s$, i.e., $v_{is} = p(\nu(\boldsymbol{x}_i, y_i) = s)$. To accelerate the optimization of $V$, we further propose a diversity regularization term $\text{Div}(\boldsymbol{x}) = \sum_{s_1, s_2 \in \mathcal{S}, s_1 \neq s_2} \|f_{s_1}(\boldsymbol{x}) - f_{s_2}(\boldsymbol{x})\|_2$, which prevents the collapse together of different subpopulations. Increasing the diversity among the outputs can also force the model to learn richer features to help prediction (Brown et al., 2005; Krogh & Vedelsby, 1994; Tang et al., 2006). Thus, the final loss function of our method can be formulated as follows:

$$\mathcal{L} = -\frac{1}{N} \sum_{i=1}^{N} \sum_{s \in \mathcal{S}} v_{is} \cdot \log f_s^{y_i}(\boldsymbol{x}_i) - \hat{H}_{\mathcal{D}}(Y|V) - \beta \frac{1}{N} \sum_{i=1}^{N} \text{Div}(\boldsymbol{x}_i) \tag{2}$$

where $\beta$ is a hyperparameter that controls the weight of the diversity regularization term.

**Multi-head strategy.** A classical classification model $f$ parameterized by $\theta$ consists of a deep feature extractor $\psi$ and a linear classifier $g$ with the parameter matrix $W$. The final prediction is denoted as $f(\boldsymbol{x}) = \text{softmax}(z(\boldsymbol{x}))$, where $z$ is the output logits of $\boldsymbol{x}$, i.e., $z(\boldsymbol{x}) = g(\psi(\boldsymbol{x})) = W^\top \psi(\boldsymbol{x})$. Since we need to obtain separate prediction results for each subpopulation in Eq. (2), we apply a multi-head strategy following Tang et al. (2020); Vaswani et al. (2017). Specifically, we equally divide the channels of the feature and the classifier weight into $K$ groups, i.e., $\psi(\boldsymbol{x}) = [\psi_1(\boldsymbol{x}), \psi_2(\boldsymbol{x}), \ldots, \psi_K(\boldsymbol{x})]$, $W = [W_1, W_2, \ldots, W_K]$ and the outputs logits for any subpopulation $s \in \mathcal{S}$ is denoted as $z_s(\boldsymbol{x}) = W_s^\top \psi_s(\boldsymbol{x})$. Thus the final subpopulation-balanced prediction is obtained by $f(\boldsymbol{x}) = \text{softmax}(z(\boldsymbol{x}))$, where $z(\boldsymbol{x}) = \log \sum_{s \in \mathcal{S}} e^{z_s(\boldsymbol{x})}$ according to Thm. 3.4. Note that, our multi-head strategy *does not introduce any additional parameters* to the network compared with the network counterpart without considering the subpopulation imbalance. That is to say, we just split the output features of the penultimate layer and the classifier weights of the last layer into different groups, and use them to generate the corresponding predictions for multiple subpopulations.

## 4 EXPERIMENT

### 4.1 EXPERIMENTAL SETUP

**Datasets.** We evaluate our SHE on COCO (Lin et al., 2014), CIFAR-100 (Krizhevsky et al., 2009), and tieredImageNet (Ren et al., 2018). For COCO, we follow the ALT-protocol (Tang et al., 2022) to conduct subpopulation-imbalanced training set and balanced test set. For CIFAR-100, we take the 20 superclasses as classification targets and generate subpopulation imbalances by sampling in the subclasses of each superclass. Following Cui et al. (2019), we use the exponential sampling with imbalance ratio IR $\in \{20, 50, 100\}$, where IR $= \frac{\max_{s \in \mathcal{S}} \sum_{(\boldsymbol{x}_i, y_i, s_i) \in \mathcal{D}} \mathbf{1}(s_i = s)}{\min_{s \in \mathcal{S}} \sum_{(\boldsymbol{x}_i, y_i, s_i) \in \mathcal{D}} \mathbf{1}(s_i = s)}$. For tieredImageNet, we take the 34 superclasses as classification targets and generate subpopulation imbalances by imbalanced sampling in 10 subclasses of each superclass with the imbalance ratio IR $= 100$.

**Baselines.** We consider extensive baselines: 1) empirical risk minimization (ERM); 2) imbalanced learning methods: PaCO (Cui et al., 2021), BCL (Zhu et al., 2022), IFL (Tang et al., 2022), DB (Ma

Table 2: Performance (Mean $\pm$ Std) of methods on COCO, CIFAR-100 with the imbalance ratio IR $\in \{100, 50, 20\}$ (marked as CIFAR-IRIR), and tieredImageNet. Bold indicates the best results.

| Method | COCO | CIFAR-IR100 | CIFAR-IR50 | CIFAR-IR20 | tieredImageNet |
|--------|------|-------------|------------|------------|----------------|
| ERM | 62.52 ± 0.32% | 52.49 ± 0.27% | 55.20 ± 0.41% | 58.92 ± 0.62% | 48.23 ± 0.27% |
| PaCO | 62.59 ± 0.24% | 52.89 ± 0.39% | 55.47 ± 0.29% | 59.15 ± 0.44% | 48.72 ± 0.45% |
| BCL | 62.83 ± 0.42% | 53.02 ± 0.26% | 55.50 ± 0.33% | 59.07 ± 0.23% | 48.56 ± 0.61% |
| IFL | 62.57 ± 0.15% | 52.45 ± 0.33% | 55.16 ± 0.42% | 59.07 ± 0.51% | 48.64 ± 0.18% |
| DB | 62.72 ± 0.48% | 52.96 ± 0.21% | 55.52 ± 0.27% | 59.19 ± 0.37% | 48.52 ± 0.13% |
| TDE | 62.64 ± 0.27% | 52.67 ± 0.12% | 55.34 ± 0.17% | 59.10 ± 0.22% | 48.36 ± 0.54% |
| ETF-DR | 62.45 ± 0.37% | 52.43 ± 0.18% | 55.27 ± 0.13% | 58.87 ± 0.17% | 48.51 ± 0.66% |
| LfF | 62.06 ± 0.83% | 52.13 ± 0.52% | 54.78 ± 0.64% | 58.54 ± 0.52% | 47.87 ± 0.23% |
| Focal | 61.67 ± 0.53% | 51.77 ± 0.63% | 54.64 ± 0.62% | 58.33 ± 0.73% | 47.68 ± 0.62% |
| EIIL | 62.61 ± 0.33% | 52.82 ± 0.17% | 55.55 ± 0.32% | 59.02 ± 0.35% | 48.56 ± 0.33% |
| ARL | 62.48 ± 0.22% | 52.67 ± 0.36% | 55.32 ± 0.17% | 59.03 ± 0.24% | 48.55 ± 0.38% |
| GRASP | 62.73 ± 0.25% | 52.92 ± 0.41% | 55.62 ± 0.30% | 59.12 ± 0.27% | 48.37 ± 0.24% |
| JTT | 62.32 ± 0.75% | 52.37 ± 0.48% | 55.02 ± 0.32% | 58.61 ± 0.64% | 48.04 ± 0.39% |
| MaskTune | 60.23 ± 0.73% | 51.63 ± 0.31% | 54.35 ± 0.49% | 58.03 ± 0.36% | 47.56 ± 0.54% |
| **SHE** | **64.56 ± 0.24%** | **54.52 ± 0.35%** | **56.87 ± 0.17%** | **60.72 ± 0.41%** | **50.14 ± 0.18%** |

et al., 2023b), TDE (Tang et al., 2020), and ETF-DR (Yang et al., 2022); 3) methods for spurious correlations that *do not require subpopulation annotation on the training and validation set*: LfF (Nam et al., 2020), Focal (Lin et al., 2017), EIIL (Creager et al., 2021), ARL (Lahoti et al., 2020), GRASP (Zeng et al., 2022), JTT (Liu et al., 2021), and MaskTune (Taghanaki et al., 2022). *Note that,* some imbalance learning methods like LA (Menon et al., 2021), LDAM (Cao et al., 2019), and CB (Cui et al., 2019) will degrade to the ERM performance when the class level is balanced.

**Implementation details.** We use 18-layer ResNet as the backbone. The standard data augmentations are applied as in Cubuk et al. (2020). The mini-batch size is set to 256 and all the methods are trained using SGD with momentum of 0.9 and weight decay of 0.005 as the optimizer. The pre-defined $K$ is set to 4 if not specifically stated and the hyper-parameter $\beta$ in Eq. (2) is set to 1.0. The initial learning rate is set to 0.1. We train the model for 200 epochs with the cosine learning-rate scheduling.

## 4.2 Performance Evaluation on Subpopulation Imbalance

**Overall performance.** In Tab. 2, we summarize the top-1 test accuracies on three datasets, COCO, CIFAR-100 with imbalance ratio IR $= \{100, 50, 20\}$ and tieredImageNet. As can be seen, SHE achieves consistent improvement over all baselines on these benchmark settings. Specifically, we achieve the gains 1.72% on COCO, 1.50%, 1.35%, 1.53% on CIFAR-100 with three imbalance ratios, and 1.42% on tieredImageNet compared to the best baseline. In comparision, imbalanced baselines usually show marginal improvement or perform comparably with ERM, whose gains mainly come from contrastive representation learning (*e.g.*, PaCO), invariant representation learning (*e.g.*, IFL), and robust classifier design (*e.g.*, ETF-DR), etc. The baselines regarding spurious correlations, on the other hand, usually assume that the model tends to fit spurious correlations, leading to performance degradation when there are no obvious spurious correlations captured by the model during training.

**Many/Medium/Few analysis.** In Tab. 3, we show the fine-grained per-split accuracies of different methods on COCO. Note that, the Many/Medium/Few three splits correspond to the training sample number of the subpopulation that ranks in the top, middle and bottom partitions. As expected, baselines generally have higher accuracy in dominant subpopulations but perform poorly in tails. On the Few-split, a gap of 4.42% is achieved between SHE and the best baseline, and we achieve the best results on Many-split and Medium-split. This shows a merit of SHE that enhances the performance of minority subpopulations without sacrificing the performance of head subpopulations.

## 4.3 Performance Evaluation on Richer Imbalance Contexts

**Training under subpopulation imbalance coupled with class imbalance.** It is practical to see how SHE performs when both class and subpopulation imbalances coexist in the data. To verify this, we follow (Tang et al., 2022) to construct a class and subpopulation imbalanced training set. For CIFAR and tieredImageNet, we construct the training set by imbalanced sampling with an imbalance ratio

Table 3: Per-split accuracies on COCO. Many, Medium, and Few are the three splits of the test set based on the training imbalancedness. Overall means the full test set. MT: the short for MaskTune. The complete experimental results (Mean ± Std) of all baselines can be found in Appx. F.3.

| Method | ERM | PaCO | BCL | IFL | DB | TDE | EIIL | ARL | GRASP | JTT | MT | **SHE** |
|---|---|---|---|---|---|---|---|---|---|---|---|---|
| Many | 67.21% | 67.45% | 66.89% | **67.71%** | 67.35% | 66.32% | 66.87% | 67.32% | 67.13% | 66.93% | 64.48% | **67.71%** |
| Medium | 52.22% | 53.33% | 53.21% | 52.17% | 52.11% | 53.23% | 52.79% | 53.34% | 53.26% | 51.24% | 50.11% | **53.50%** |
| Few | 37.10% | 36.23% | 37.67% | 36.82% | 37.47% | 37.02% | 37.06% | 37.18% | 37.29% | 36.48% | 33.27% | **42.09%** |
| Overall | 62.52% | 62.59% | 62.93% | 62.57% | 62.72% | 62.64% | 62.61% | 62.48% | 62.73% | 62.32% | 60.23% | **64.56%** |

Table 4: Performance under more imbalance settings. Bold indicates superior results. (Left) Performance on COCO, CIFAR-100 (IR = 100), and tieredImageNet where both class imbalance and subpopulation imbalance co-exist (Mean ± Std). (Right) Performance on datasets for spurious correlations. The worst group accuracy (Worst Acc) and the average accuracy (Mean Acc) is reported. The second column means whether using the group annotation on the training or validation set.

| Setting: both subpopulation and class imbalance (Mean ± Std) | | | | Setting: spurious correlation (Worst Acc / Mean Acc) | | | |
|---|---|---|---|---|---|---|---|
| Method | COCO | CIFAR-IR100 | tieredImageNet | Method | Group Info (Train / Val) | CelebA | Waterbirds |
| ERM | 63.57 ± 0.34% | 59.24 ± 0.46% | 53.65 ± 0.46% | | | | |
| LA | 66.47 ± 0.27% | 59.73 ± 0.27% | 54.12 ± 0.35% | GDRO | Yes / Yes | **88.3%** / 91.8% | **91.4%** / 93.5% |
| LDAM | 66.32 ± 0.33% | 59.66 ± 0.26% | 54.01 ± 0.51% | LfF | No / Yes | 77.2% / 85.1% | 82.1% / 94.3% |
| CB | 66.17 ± 0.21% | 59.45 ± 0.36% | 53.78 ± 0.21% | SD | No / Yes | **83.2%** / 91.6% | **87.3%** / 90.3% |
| PaCO | 66.78 ± 0.41% | 59.87 ± 0.51% | 54.15 ± 0.39% | JTT | No / Yes | 81.1% / 88.0% | 86.7% / 93.3% |
| BCL | 66.92 ± 0.26% | 59.78 ± 0.37% | 54.23 ± 0.27% | CIM | No / Yes | 81.3% / 89.2% | 77.2% / 95.6% |
| IFL | 65.34 ± 0.52% | 59.44 ± 0.41% | 53.88 ± 0.43% | ERM | No / No | 47.2% / 95.6% | 74.9% / 98.1% |
| DB | 66.43 ± 0.15% | 59.81 ± 0.29% | 54.14 ± 0.30% | LfF | No / No | 24.4% / 85.1% | 67.5% / 87.5% |
| TDE | 66.12 ± 0.44% | 59.63 ± 0.34% | 53.91 ± 0.28% | JTT | No / No | 40.6% / 88.0% | 71.8% / 92.3% |
| ETF-DR | 65.92 ± 0.26% | 59.71 ± 0.18% | 54.07 ± 0.31% | MaskTune | No / No | **78.0%** / 91.3% | 80.7% / 92.1% |
| **SHE$_{w/LA}$** | **68.11 ± 0.27%** | **61.67 ± 0.31%** | **55.73 ± 0.22%** | SHE$_{w/GDRO}$ | No / No | 77.9% / 91.7% | **81.9%** / 91.3% |

IR = 100 on both classes and subpopulations. The classes and subpopulations are both balanced on the test set. According to the results in Tab. 4 (left), we can see that the imbalance learning baselines consistently improve test accuracy compared to ERM when class imbalance also exists. When we combine SHE with a classical imbalanced learning baseline LA (Menon et al., 2021), our SHE$_{w/LA}$ achieves a 1.19% improvement on COCO, 1.80% on CIFAR and 1.50% on tieredImageNet compared to the best baseline, showing the potential of SHE on more complex imbalance learning problems.

**Training under spurious correlations.** We directly apply SHE into GDRO (Sagawa et al., 2019) (using the learned subpopulations instead of the prior subgroup annotations) to verify the effectiveness on spurious correlation datasets, CelebA (Liu et al., 2015) and Waterbirds (Sagawa et al., 2019). In Tab. 4 (right), we compare SHE$_{w/GDRO}$ with a series of baselines, and our method achieves the promising performance in mitigating spurious correlations when there is no group information available. Methods that require group annotations (*e.g.*, SD (Pezeshki et al., 2021) and CIM (Taghanaki et al., 2021)) are also exhibited for reference. Interestingly, more visualization results in Appx. F.2 show that the performance comes from dividing the training data into two meaningful subpopulations: data *w/* and *w/o* spurious correlations, which is actually different from the prior group annotations.

### 4.4 ABLATION STUDY AND ANALYSIS

**Ablation on varying the latent subpopulation number $K$.** To study the effect of the latent subpopulation number $K$ in SHE, we conduct ablation on COCO as shown in Fig. 4(a). When $K = 1$, Eq. (2) degenerates to the cross-entropy loss, and so is performance. When $K > 1$, SHE shows a significant improvement over ERM and is robust to $K$. At $K = 4$, our SHE achieves the best results on average. Similar phenomenon on CIFAR and tieredImageNet can be found in Appx. F.5.

**Effect of (a) the diversity term and (b) the entropy term.** To study the effect of the diversity term $\text{Div}(x)$ in Eq. (2), we conduct experiments on $\beta$ on COCO. As shown in Fig. 4(b), even without the diversity term ($\beta = 0$), SHE still significantly outperforms the ERM baseline. The addition of the diversity term continually enhances the performance to the best on average at $\beta = 1.0$, and SHE is generally robust to the choice of $\beta$. We also conduct a comparison with SHE without the entropy term $H_{\mathcal{D}}(Y|V)$ in Eq. (2) (termed as SHE$_{w/o\ entropy}$) in Tab. 5, which confirms that the entropy term consistently and effectively enhances the performance.

**Effect of pursuing (a) the optimal data partition and (b) the subpopulation-balanced prediction.** In Tab. 5, we present the performance of ERM, ERM with the multi-head strategy (namely

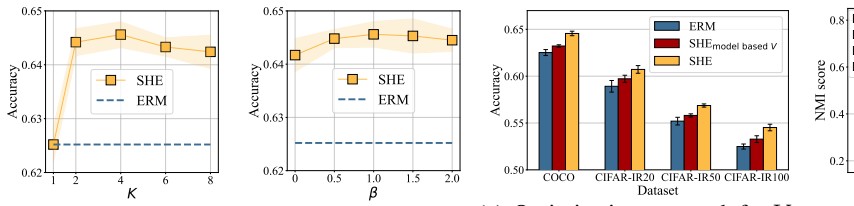

(a) Ablation on $K$    (b) Ablation on $\beta$    (c) Optimization approach for $V$.    (d) NMI score

Figure 4: (a) Performance of SHE and ERM on COCO with varying subpopulation number $K$. (b) Performance of SHE and ERM on COCO with varying $\beta$. (c) Performance of ERM, SHE$_{\text{model based }V}$, and SHE on COCO, CIFAR-IR20, CIFAR-IR50, and CIFAR-IR100. (d) NMI scores between the learned subpopulations and the true annotations on Waterbird.

Table 5: Performance of ERM, SHE, and some of their variants on COCO and CIFAR-100.

| Method | ERM | ERM$_{\text{multi-head}}$ | SHE$_{\text{EIIL}}$ | SHE$_{\text{w/o entropy}}$ | **SHE** |
|---|---|---|---|---|---|
| COCO | $62.52 \pm 0.32\%$ | $62.47 \pm 0.28\%$ | $62.82 \pm 0.27\%$ | $64.15 \pm 0.27\%$ | $\mathbf{64.56 \pm 0.24\%}$ |
| CIFAR-IR100 | $52.49 \pm 0.27\%$ | $52.53 \pm 0.17\%$ | $52.63 \pm 0.22\%$ | $53.96 \pm 0.37\%$ | $\mathbf{54.52 \pm 0.35\%}$ |
| CIFAR-IR50 | $55.20 \pm 0.41\%$ | $55.16 \pm 0.47\%$ | $55.36 \pm 0.37\%$ | $56.31 \pm 0.23\%$ | $\mathbf{56.87 \pm 0.17\%}$ |
| CIFAR-IR20 | $58.92 \pm 0.62\%$ | $58.88 \pm 0.36\%$ | $59.21 \pm 0.48\%$ | $60.03 \pm 0.38\%$ | $\mathbf{60.72 \pm 0.41\%}$ |

ERM$_{\text{multi-head}}$), and SHE by removing the multi-head network but following the way of EIIL to utilize the learned subpopulations (namely SHE$_{\text{EIIL}}$). SHE achieves a significant improvement over ERM and ERM$_{\text{multi-head}}$, while ERM$_{\text{multi-head}}$ achieves only comparable results to ERM, showing the necessity of pursuing the optimal data partition. The component of SHE to pursue subpopulation-balanced predictions is better (SHE vs. SHE$_{\text{EIIL}}$), which confirms its effectiveness.

**Analysis on the optimization approach for subpopulation-weight matrix $V$.** We construct a variant of SHE uses a model-based approach to learn the data partition from image features, namely SHE$_{\text{model based }V}$. As can be seen in Fig. 4(c), SHE$_{\text{model based }V}$ shows a clear performance degradation compared to SHE. A possible reason is that $\nu$ in Def. 3.1 is a function of both the input $x$ and the label $y$, but SHE$_{\text{model based }V}$ can only learn the data partition from $x$.

**Quality of the recovered subpopulations.** To investigate the capability of SHE in discovering subpopulations in training data, we conduct a comparative analysis between SHE and baselines based on subgroup inference (EIIL, ARL, GRASP). Specifically, Fig. 4(d) presents the NMI scores on Waterbird between the recovered subpopulations and the ground truth annotations. Our SHE exhibits a remarkable capability to accurately discover the latent structures within the training data.

**Fine-tuning from pre-trained models.** Foundation models have achieved impressive performance in numerous areas in recent years (Radford et al., 2021; Rogers et al., 2020; Brown et al., 2020). Fine-tuning from these pre-trained models using downstream training data is gradually becoming a prevalent paradigm. In Tab. 6, we exhibit the results of different methods fine-tuned on the COCO dataset with three multimodal pre-training models, *i.e.*, CLIP (ViT-B/32) (Radford et al., 2021), ALIGN (EfficientNet-L2 & BERT-Large) (Jia et al., 2021), and AltCLIP (ViT-L) (Chen et al., 2022). The LoRA (Hu et al., 2022) technique is used for fine-tuning to speed up training and prevent overfitting. Despite the notable improvements obtained through fine-tuning compared to training from scratch, SHE consistently surpasses all baselines with different large-scale pre-trained models.

Table 6: LoRA fine-tuning of different methods under three popular pre-trained models on COCO. The complete results (Mean $\pm$ Std) can be found in Appx. F.4.

| Method | CLIP | ALIGN | AltCLIP |
|---|---|---|---|
| Zero-shot | 76.59% | 78.45% | 82.55% |
| ERM | 84.46% | 83.23% | 84.93% |
| BCL | 84.43% | 83.42% | 85.01% |
| IFL | 84.49% | 83.36% | 84.89% |
| LfF | 84.27% | 83.05% | 84.17% |
| JTT | 84.37% | 83.07% | 84.55% |
| MaskTune | 83.37% | 82.66% | 83.92% |
| **SHE** | **85.34%** | **84.19%** | **85.76%** |

## 5 CONCLUSION

In this paper, we focus on a hidden subpopulation imbalance scenario and identify its several critical challenges. To alleviate the subpopulation imbalance problem, we first introduce the concept of optimal data partition, which splits the data into the subpopulations that are most helpful for prediction. Then, a novel method, SHE, is proposed to uncover and balance hidden subpopulations in training data during training. It is theoretically demonstrated that our method converges to optimal data partition and makes balanced predictions. Empirical evidence likewise demonstrates that our method uncovers meaningful latent structures in the data. Extensive experiments under diverse settings and different configures consistently demonstrate the effectiveness of SHE over a range of baselines.

ETHICS STATEMENT

By discovering the latent subpopulations in the training data and encouraging subpopulation-balanced predictions, the paper aims to improve the generalization and performance parity of machine learning models across different subpopulations of data. This can have important implications for various social applications, such as medical diagnosis, auto-driving, and criminal justice, where subpopulation imbalance may exist and lead to biased or inaccurate outcomes. Our proposed method does not require the annotation of subpopulation or even the predefined semantics of subpopulation, which reduces the cost of data annotation and on the other hand avoids the serious fairness consequences of annotation omission. Negative impacts may also occur when the proposed subpopulation discovery technology falls into the wrong hands, for example, it can be used to identify minorities for malicious purposes. Therefore, it is the responsibility to ensure that such technologies are used for the right purposes.

REPRODUCIBILITY STATEMENT

All the experiments are conducted on NVIDIA GeForce RTX 3090s with Python 3.7.10 and Pytorch 1.13.1. We provide experimental setups and implementation details in Sec. 4.1 and Appx. F.1. The theoretical proofs are given in Appx. C.

ACKNOWLEDGEMENT

This work is supported by the National Key R&D Program of China (No. 2022ZD0160702), STCSM (No. 22511106101, No. 22511105700, No. 21DZ1100100), 111 plan (No. BP0719010) and National Natural Science Foundation of China (No. 62306178).

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

# APPENDIX

## CONTENTS

# A   BROADER RELATED WORK

## A.1   INFORMATION THEORY-GUIDED OBJECTIVE DESIGN

**Information Bottleneck.** Information Bottleneck Theory (Tishby et al., 2000; Slonim & Tishby, 2000) in deep learning is a concept that has been extensively researched and developed over the years. It focuses on optimizing neural networks by maximizing the relevant information about the target while minimizing redundant data. The information bottleneck has gained widespread attention in recent years within the field of deep learning (Tishby & Zaslavsky, 2015; Goldfeld & Polyanskiy, 2020; Kawaguchi et al., 2023). For instance, gradient-based methods were employed in optimizing a Deep Neural Network (DNN) to tackle the Information Bottleneck Lagrangian (Alemi et al., 2017). This approach, known as the deep variational IB (VIB), enables the system to learn stochastic representation rules, showcasing enhanced generalization capabilities and robustness to adversarial examples. A similar objective was explored in Achille & Soatto (2018), where the emphasis was on promoting minimality, sufficiency, and disentanglement of representations. This disentanglement property was also harnessed for generative modeling purposes, leading to the development of the $\beta$-variational autoencoder (Higgins et al., 2017).

**Mutual Information Maximization**:   Mutual Information Maximization (InfoMax) principle (Linsker, 1988) is a common training objective designed to enhance the information sharing between model outputs and target variables. This approach is particularly popular in self-supervised learning and representation learning and have demonstrated promising empirical results (Hjelm et al., 2019; Tschannen et al., 2020; Chen et al., 2020; 2021; 2023; Zhao et al., 2024). In general, their objective is to maximize the mutual information between representations from different views of the same image. For instance, in DeepInfoMax (Hjelm et al., 2019), $g_1$ extracts overall features from the entire image, and $g_2$ captures local features from patches, where $g_1$ and $g_2$ are activations in different layers of the same convolutional network. Extending this idea, Bachman et al. (2019) generate the two views by using different augmentations of the same image. Contrastive Multiview Coding (Tian et al., 2020) extends the objective to incorporate multiple views, with each view corresponding to a different image modality.

**Comparision with our work.** The information bottleneck and mutual information maximization techniques involve the mutual information between input, representation, and label variables, aiming to optimize the network for learning effective classifiers or generalizable representations. In contrast to these methods, our method, distinctively, models mutual information maximization (Def. 3.1) with the direct purpose of learning an effective data partition, which further serves the subpopulation harmonization.

## A.2   DOMAIN GENERALIZATION

The objective of domain generalization is to extract knowledge that is invariant across diverse source domains and generalize it to novel, unseen target domains. A multitude of methods have emerged for domain generalization, broadly categorized into five groups: domain alignment, meta learning, domain hallucination, architecture-based methods, and regularization-based methods (Xu et al., 2023). Domain alignment methods (Li et al., 2018; Zhao et al., 2020b; Grill et al., 2020; Ye et al., 2023) target on minimizing the discrepancies between source domains to learn domain-invariant features. Meta-learning-based methods (Balaji et al., 2018; Li et al., 2019; Zhang et al., 2023a) enhance the generalizability of models to domain shifts by partitioning training domains into distinct meta-train and meta-test domains. Through meta-optimization, these methods simulate unseen domain shifts, thereby improving the models' adaptability to such shifts. Domain hallucination (Zhou et al., 2021; Xu et al., 2021) aims to augment training samples by transforming the original samples into specific unseen domains while preserving their underlying semantics. Architecture-based methods (Chattopadhyay et al., 2020; Chang et al., 2019) typically involve designing domain-specific modules for different domains. During final testing, these methods aggregate results inferred from all source domains to achieve a comprehensive outcome. Regularization-based domain generalization methods (Shi et al., 2020; Carlucci et al., 2019) involve learning general and universal features across domains through various regularization techniques.

**Comparision with our work.** The main distinctions between domain generalization and the sub-population imbalance problem discussed in our paper are as follows: (1) In domain generalization,

domain labels are accessible during training, whereas subpopulation annotations are not visible. (2) The goal of domain generalization is to exhibit strong generalization performance on unseen domains, while the problem of subpopulation imbalance aims for a comprehensive performance across all encountered subpopulations. Furthermore, the concepts of domain and subpopulation differ in that domains are more akin to image styles unrelated to semantics, while subpopulations represent a kind of semantic abstraction distinct from the class dimension. Therefore, domain generalization methods typically aim to learn domain-invariant features. In contrast, our method separates the learning of subpopulations with different prediction mechanisms and then balancing them.

### A.3  Algorithmic Fairness

Fairness is a critical and extensively studied aspect in algorithmic decision-making. When dealing with biased data, algorithms tend to make decisions based on attributes that is sensitive or should be protected(*e.g.*, race and gender), raising concerns about fairness (Kearns et al., 2018). Various approaches in algorithmic fairness aim to mitigate this issue by introducing fair constraints during the training procedure, such as demographic parity or equalized odds (Hardt et al., 2016; Jiang et al., 2019; Calmon et al., 2017). Additionally, alternative fairness criteria include accuracy parity (Zhao et al., 2020a; Sagawa et al., 2020) (ensuring uniform accuracy across subgroups), small prediction variance  (Li et al., 2020; 2021) (maintaining minimal prediction variations among subgroups) and small prediction loss for all subgroups (Zafar et al., 2019; Hashimoto et al., 2018). Some work introduces independence constraints to the objective to ensure that decisions do not rely on sensitive attributes (Madras et al., 2018; Song et al., 2019).

**Comparision with our work.** In the algorithmic fairness problem, the protected attribute annotations are sometimes visible and sometimes not. When attribute annotations are invisible, the algorithmic fairness problem bears some similarities to our problem. The key difference is that algorithmic fairness often aims to protect specific sensitive attributes, preventing the model from using them in decision-making. In our case, different subpopulations have different decision mechanisms, and we want to preserve features of minority subpopulations. In essence, while algorithmic fairness tends to learn fewer features (excluding protected attributes), we aim to learn more features (safeguarding features of the disadvantaged subpopulations). Additionally, the evaluation metrics differ between the algorithmic fairness and our problem.

### A.4  Comparision with Creager et al. (2021) and Lahoti et al. (2020)

EIIL(Creager et al., 2021) and ARL(Lahoti et al., 2020) infer subgroup membership based on the violation degree under the invariant learning principle or the stability of the loss space, and then perform group-invariant learning or reweighting. In comparison, SHE aims to optimize the predictive ability, specifically the interaction information, by partitioning data into subpopulations and concurrently rebalancing predictions among these subpopulations. Besides, these works assume that the subpopulation distribution is causally independent of the predicted target, which may not always hold in our scenario.

## B  Supplementary Equations

### B.1  Mutual Information and Entropy

**Mutual Information.** In probability theory and information theory, the mutual information (MI) of two random variables is a measure of the mutual dependence between the two variables. The mutual information of two jointly random variables $X$ (continuous) and $Y$ (discrete) is defined as

$$I(X;Y) = \sum_{y \in \mathcal{Y}} \int_{\mathcal{X}} p(\boldsymbol{x}, y) \log(\frac{p(\boldsymbol{x}, y)}{p(\boldsymbol{x})p(y)}) d\boldsymbol{x}. \tag{3}$$

**Relation to entropy.** Mutual information can be equivalently expressed as

$$I(X;Y) = H(Y) - H(Y|X) = H(X) - H(X|Y), \tag{4}$$

where $H(Y)$ and $H(X)$ are (marginal) entropies, and $H(Y|X)$ and $H(X|Y)$ are conditional entropies.

**Entropy.** The entropy $H$ for a discrete continuous variable $X$ and a continuous variable $Y$ can be defined as follows, respectively.

$$H(X) = -\int_{\mathcal{X}} p(\boldsymbol{x}) \log(p(\boldsymbol{x})) d\boldsymbol{x} \tag{5}$$

$$H(Y) = -\sum_{y \in \mathcal{Y}} p(y) \log(p(y)) \tag{6}$$

**Empirical Entropy.** Here we provide the equation for the empirical entropy in Eq. (1):

$$\hat{H}_{\mathcal{D}}(Y) = \sum_{y \in \mathcal{Y}} \pi_{\mathcal{D}}(y) \log \pi_{\mathcal{D}}(y),$$

$$\hat{H}_{\mathcal{D}}(Y|\nu(X,Y)) = \sum_{s \in \mathcal{S}} \pi_{\mathcal{D}}(s) \sum_{y \in \mathcal{Y}} \pi_{\mathcal{D}}(y|s) \log \pi_{\mathcal{D}}(y|s), \tag{7}$$

where $\pi_{\mathcal{D}}(y) = \frac{1}{N} \sum_{i=1}^{N} 1(y_i = y)$, $\pi_{\mathcal{D}}(s) = \frac{1}{N} \sum_{i=1}^{N} 1(\nu(x_i, y_i) = s)$, $\pi_{\mathcal{D}}(y|s) = \frac{\sum_{i=1}^{N} 1(y_i=y)1(\nu(x_i,y_i)=s)}{\sum_{i=1}^{N} 1(\nu(x_i,y_i)=s)}$ are empirical frequencies in the training set.

## B.2 NORMALIZED MUTUAL INFORMATION (NMI) SCORE

The Normalized Mutual Information (NMI) (Strehl & Ghosh, 2002; Li et al., 2024; 2023) between the learned data partition $\nu$ (for simplicity here we will use $\nu$ to denote $\nu(X,Y)$.) and the ground truth subpopulation $S$ is defined as:

$$\text{NMI}(S, \nu) = \frac{2I(S; \nu)}{H(S) + H(\nu)}, \tag{8}$$

which is a normalization of the Mutual Information (MI) score to scale the results between $0$ (no correlation) and $1$ (perfect correlation). We use the empirical score of NMI in our experiments.

## B.3 RADEMACHER COMPLEXITY IN THM. 3.3

The Rademacher complexity (Bartlett & Mendelson, 2002) is a concept used in statistical learning theory and machine learning to measure the complexity of a class of functions. It provides a way to quantify how well a function class can fit random noise, which in turn helps in understanding the capacity of the class to overfit training data.

Consider a sample space $\mathcal{X} \times \mathcal{Y}$ and a class of real-valued functions $G$ defined on $\mathcal{X} \times \mathcal{Y} \to \mathbb{R}$. For a sample set $\mathcal{D}' = \{(\boldsymbol{x}_1, y_1), (\boldsymbol{x}_2, y_2), ..., (\boldsymbol{x}_N, y_N)\}$ drawn i.i.d. from $\mathcal{X} \times \mathcal{Y}$, introduce Rademacher random variables $\sigma_1, \sigma_2, ..., \sigma_N$, which are independent and take values +1 or -1 with equal probability 1/2.

The empirical Rademacher complexity of the function class $G$ with respect to the sample is defined as the expected value of the supremum (maximum) of the average sum of the product of $g(\boldsymbol{x}_i, y_i)$ and $\sigma_i$ over all functions $g$ in $G$. Mathematically, it's expressed as:

$$\hat{R}_N(G) = \mathbb{E}_\sigma \left[ \sup_{g \in G} \frac{1}{N} \sum_{i=1}^{N} \sigma_i g(\boldsymbol{x}_i, y_i) \right] \tag{9}$$

The Rademacher complexity $R_N(G)$ of the class $G$ is the expectation of the empirical Rademacher complexity over all sample set of size $N$ drawn from the space $\mathcal{X} \times \mathcal{Y}$.

$$R_N(G) = \mathbb{E}_{\mathcal{D}'} \left[ \hat{R}_N(G) \right] \tag{10}$$

### B.4 McDiarmid's Inequality

McDiarmid's Inequality (McDiarmid et al., 1989) is a concentration inequality which provides bounds on the probability that a function of independent random variables deviates significantly from its expected value.

Let $X_1, X_2, \ldots, X_n$ be independent random variables taking values in $\mathcal{X}$. Consider a function $f : \mathcal{X}^N \to \mathbb{R}$ satisfying the following bounded difference condition:

For each $i \in \{1, 2, \ldots, n\}$ and for any $x_1, \ldots, x_n, x_i' \in X_n$,

$$\left| f(x_1, \ldots, x_{i-1}, x_i, x_{i+1}, \ldots, x_n) - f(x_1, \ldots, x_{i-1}, x_i', x_{i+1}, \ldots, x_n) \right| \leq c_i \tag{11}$$

where $c_i$ is a constant.

Then, for any $\epsilon > 0$,

$$\Pr\left[ f(X_1, X_2, \ldots, X_n) - \mathbb{E}[f(X_1, X_2, \ldots, X_n)] \geq \epsilon \right] \leq \exp\left( -\frac{2\epsilon^2}{\sum_{i=1}^n c_i^2} \right) \tag{12}$$

and similarly,

$$\Pr\left[ f(X_1, X_2, \ldots, X_n) - \mathbb{E}[f(X_1, X_2, \ldots, X_n)] \leq -\epsilon \right] \leq \exp\left( -\frac{2\epsilon^2}{\sum_{i=1}^n c_i^2} \right) \tag{13}$$

This inequality is particularly useful in scenarios where one wishes to control the deviations of a function of several independent variables from its expected value, especially in the context of machine learning and statistical learning theory.

## C Theoretical Proofs

### C.1 Proof of Prop. 3.2

*Proof of Prop. 3.2.*
For a data partition $\nu'$ that satisfies $\nu' \perp\!\!\!\perp X, Y$, we have

$$I(X; Y | \nu'(X, Y)) = I(X; Y) \tag{14}$$

For the optimal data partition $\nu^*$, according to Def. 3.1, we have

$$I(X; Y; \nu^*(X, Y)) \geq I(X; Y; \nu'(X, Y)) = 0 \tag{15}$$

$\square$

### C.2 Proof of Thm. 3.3

**Lemma C.1.** $H(Y|X) = \inf_f \mathbb{E}_{\boldsymbol{x},y} - \log f^y(\boldsymbol{x})$, *where the infimum is achieved when* $f^y(\boldsymbol{x}) = p(y|\boldsymbol{x})$.

Proof for Lem. C.1 follows the same strategy as Proposition 1 in Xu et al. (2020).

*Proof of Lem. C.1.*

$$\begin{aligned}
&\inf_f \mathbb{E}_{\boldsymbol{x},y} - \log f^y(\boldsymbol{x}) \\
&= \inf_f \mathbb{E}_{\boldsymbol{x}} \mathbb{E}_{y|\boldsymbol{x}} \log \frac{p(y|\boldsymbol{x})}{f^y(\boldsymbol{x})p(y|\boldsymbol{x})} \\
&= \inf_f \mathbb{E}_{\boldsymbol{x}} (KL(p(Y|\boldsymbol{x})||f(\boldsymbol{x})) + H(Y|\boldsymbol{x})) \\
&= \mathbb{E}_{\boldsymbol{x}} H(Y|\boldsymbol{x}) = H(Y|X),
\end{aligned} \tag{16}$$

where the infimum is achieved when $f^y(\boldsymbol{x}) = p(y|\boldsymbol{x})$.

$\square$

**Corollary C.2.**

1. $H(Y) = \inf_f \mathbb{E}_{\boldsymbol{x},y} - \log f^y(\mathbf{0})$, *where* $\mathbf{0}$ *denotes the empty input and the infimum is achieved when* $f^y(\mathbf{0}) = p(y)$.

2. $H(Y|X; \nu(X,Y)) = \inf_f \mathbb{E}_{\boldsymbol{x},y} \sum_{s \in \mathcal{S}} -1(\nu(\boldsymbol{x},y) = s) \log f_s^y(\boldsymbol{x})$, *where the infimum is achieved when* $f_s^y(\boldsymbol{x}) = p(y|\boldsymbol{x}; s)$.

3. $H(Y|\nu(X,Y)) = \inf_f \mathbb{E}_{\boldsymbol{x},y} \sum_{s \in \mathcal{S}} -1(\nu(\boldsymbol{x},y) = s) \log f_s^y(\mathbf{0})$, *where the infimum is achieved when* $f_s^y(\mathbf{0}) = p(y|s)$.

4. $\hat{H}_{\mathcal{D}}(Y|\nu(X,Y)) = \inf_f \sum_{i=1}^N \sum_{s \in \mathcal{S}} -1(\nu(\boldsymbol{x}_i,y_i) = s) \log f_s^y(\mathbf{0})$, *where the infimum is achieved when* $f_s^y(\mathbf{0}) = \frac{\sum_{i=1}^N 1(\nu(\boldsymbol{x}_i,y_i)=s)1(y_i=y)}{\sum_{i=1}^N 1(\nu(\boldsymbol{x}_i,y_i)=s)}$.

The proof of Cor. C.2 is similar to proof of Lem. C.1.

*Proof of Thm. 3.3.*
Define a function $T$ for any training set $\mathcal{D}$:

$$T(\mathcal{D}) = |(I(X;Y;\nu(X,Y))) - (-\hat{\mathcal{R}}(f^\dagger, \nu; \mathcal{D}) + B)| \tag{17}$$

According to Lem. C.1 and Cor. C.2, we have:

$$T(\mathcal{D}) = |I(X;Y|\nu(X,Y)) - I(X;Y) - \frac{1}{N} \sum_{i=1}^N \sum_{s \in \mathcal{S}} \mathbf{1}(\nu(\boldsymbol{x}_i,y_i) = s) \cdot \log f_s^{y_i}(\boldsymbol{x}_i) - \hat{H}_{\mathcal{D}}(Y|\nu(X,Y)) - B|$$

$$= |I(X;Y|\nu(X,Y)) - \frac{1}{N} \sum_{i=1}^N \sum_{s \in \mathcal{S}} \mathbf{1}(\nu(\boldsymbol{x}_i,y_i) = s) \cdot \log f_s^{\dagger y_i}(\boldsymbol{x}_i) - \hat{H}_{\mathcal{D}}(Y|\nu(X,Y))|$$

$$= |-\inf_f \mathbb{E}_{\boldsymbol{x},y} \sum_{s \in \mathcal{S}} -1(\nu(\boldsymbol{x},y) = s) \log f_s^y(\boldsymbol{x}) + \inf_f \mathbb{E}_{\boldsymbol{x},y} \sum_{s \in \mathcal{S}} -1(\nu(\boldsymbol{x},y) = s) \log f_s^y(\mathbf{0})$$

$$+ \inf_f -\frac{1}{N} \sum_{i=1}^N \sum_{s \in \mathcal{S}} \mathbf{1}(\nu(\boldsymbol{x}_i,y_i) = s) \cdot \log f_s^{y_i}(\boldsymbol{x}_i) - \inf_f \sum_{i=1}^N \sum_{s \in \mathcal{S}} -1(\nu(\boldsymbol{x}_i,y_i) = s) \log f_s^y(\mathbf{0})|$$

$$\leq \sup_f |\mathbb{E}_{\boldsymbol{x},y} \sum_{s \in \mathcal{S}} -1(\nu(\boldsymbol{x},y) = s) \log f_s^y(\boldsymbol{x}) - \frac{1}{N} \sum_{i=1}^N \sum_{s \in \mathcal{S}} \mathbf{1}(\nu(\boldsymbol{x}_i,y_i) = s) \cdot - \log f_s^{y_i}(\boldsymbol{x}_i)$$

$$- \mathbb{E}_{\boldsymbol{x},y} \sum_{s \in \mathcal{S}} 1(\nu(\boldsymbol{x},y) = s) - \log f_s^y(\mathbf{0}) + \frac{1}{N} \sum_{i=1}^N \sum_{s \in \mathcal{S}} \mathbf{1}(\nu(\boldsymbol{x}_i,y_i) = s) \cdot - \log f_s^{y_i}(\mathbf{0})|$$

$$\tag{18}$$

We define $Q(\mathcal{D})$ as

$$Q(\mathcal{D}) := \sup_f |\mathbb{E}_{\boldsymbol{x},y} \sum_{s \in \mathcal{S}} -1(\nu(\boldsymbol{x},y) = s) \log f_s^y(\boldsymbol{x}) - \frac{1}{N} \sum_{i=1}^N \sum_{s \in \mathcal{S}} \mathbf{1}(\nu(\boldsymbol{x}_i,y_i) = s) \cdot - \log f_s^{y_i}(\boldsymbol{x}_i)$$

$$- \mathbb{E}_{\boldsymbol{x},y} \sum_{s \in \mathcal{S}} 1(\nu(\boldsymbol{x},y) = s) - \log f_s^y(\mathbf{0}) + \frac{1}{N} \sum_{i=1}^N \sum_{s \in \mathcal{S}} \mathbf{1}(\nu(\boldsymbol{x}_i,y_i) = s) \cdot - \log f_s^{y_i}(\mathbf{0})|$$

$$\tag{19}$$

Let $\mathcal{D}$ and $\mathcal{D}'$ be two identical data sets except that the $j$-th data point is different.

$$Q(\mathcal{D}) - Q(\mathcal{D}')$$

$$\leq \sup_f (|\mathbb{E}_{\boldsymbol{x},y} \sum_{s \in \mathcal{S}} -\mathbf{1}(\nu(\boldsymbol{x},y) = s) \log f_s^y(\boldsymbol{x}) - \frac{1}{N} \sum_{i=1}^{N} \sum_{s \in \mathcal{S}} \mathbf{1}(\nu(\boldsymbol{x}_i,y_i) = s) \cdot - \log f_s^{y_i}(\boldsymbol{x}_i)$$

$$- \mathbb{E}_{\boldsymbol{x},y} \sum_{s \in \mathcal{S}} \mathbf{1}(\nu(\boldsymbol{x},y) = s) - \log f_s^y(\boldsymbol{0}) + \frac{1}{N} \sum_{i=1}^{N} \sum_{s \in \mathcal{S}} \mathbf{1}(\nu(\boldsymbol{x}_i,y_i) = s) \cdot - \log f_s^{y_i}(\boldsymbol{0})|$$

$$- |\mathbb{E}_{\boldsymbol{x},y} \sum_{s \in \mathcal{S}} -\mathbf{1}(\nu(\boldsymbol{x},y) = s) \log f_s^y(\boldsymbol{x}) - \frac{1}{N} \sum_{i=1}^{N} \sum_{s \in \mathcal{S}} \mathbf{1}(\nu(\boldsymbol{x}_i',y_i') = s) \cdot - \log f_s^{y_i'}(\boldsymbol{x}_i')$$

$$- \mathbb{E}_{\boldsymbol{x},y} \sum_{s \in \mathcal{S}} \mathbf{1}(\nu(\boldsymbol{x},y) = s) - \log f_s^y(\boldsymbol{0}) + \frac{1}{N} \sum_{i=1}^{N} \sum_{s \in \mathcal{S}} \mathbf{1}(\nu(\boldsymbol{x}_i',y_i') = s) \cdot - \log f_s^{y_i'}(\boldsymbol{0})|)$$

$$\leq \sup_f |\frac{1}{N} \log f_{\nu(\boldsymbol{x}_j,y_j)}^{y_j}(\boldsymbol{x}_j) - \frac{1}{N} \log f_{\nu(\boldsymbol{x}_j,y_j)}^{y_j}(\boldsymbol{0}) - \frac{1}{N} \log f_{\nu(\boldsymbol{x}_j',y_j')}^{y_j'}(\boldsymbol{x}_j') + \frac{1}{N} \log f_{\nu(\boldsymbol{x}_j',y_j')}^{y_j'}(\boldsymbol{0})|$$

$$\leq \frac{2m}{N}$$

$$(20)$$

According to McDiarmid's inequality (McDiarmid et al., 1989) (also introduced in Appx. B.4), $\forall \delta \in (0,1)$ we have:

$$Q(\mathcal{D}) \leq \mathbb{E}_{\mathcal{D}} Q(\mathcal{D}) + \frac{m}{\sqrt{N}} \sqrt{-2 \log \delta} \tag{21}$$

with probability at least $1 - \delta$.

For simplicity of writing, we define

$$A = \mathbb{E}_{\boldsymbol{x},y} \sum_{s \in \mathcal{S}} -\mathbf{1}(\nu(\boldsymbol{x},y) = s) \log f_s^y(\boldsymbol{x}) - \mathbb{E}_{\boldsymbol{x},y} \sum_{s \in \mathcal{S}} \mathbf{1}(\nu(\boldsymbol{x},y) = s) - \log f_s^y(\boldsymbol{0})$$

$$\hat{A}_{\mathcal{D}} = \frac{1}{N} \sum_{i=1}^{N} \sum_{s \in \mathcal{S}} \mathbf{1}(\nu(\boldsymbol{x}_i,y_i) = s) \cdot - \log f_s^{y_i}(\boldsymbol{x}_i) - \frac{1}{N} \sum_{i=1}^{N} \sum_{s \in \mathcal{S}} \mathbf{1}(\nu(\boldsymbol{x}_i,y_i) = s) \cdot - \log f_s^{y_i}(\boldsymbol{0})$$

$$(22)$$

So $Q(\mathcal{D}) \leq \sup_f |A - \hat{A}_{\mathcal{D}}|$, and we have:

$$\mathbb{E}_{\mathcal{D}} Q(\mathcal{D})$$

$$= \mathbb{E}_{\mathcal{D}} \sup_f |A - \hat{A}_{\mathcal{D}}|$$

$$= \mathbb{E}_{\mathcal{D}} \sup_f |\mathbb{E}_{\mathcal{D}'} \hat{A}_{\mathcal{D}'} - \hat{A}_{\mathcal{D}}|$$

$$\leq \mathbb{E}_{\mathcal{D}} \sup_f \mathbb{E}_{\mathcal{D}'} |\hat{A}_{\mathcal{D}'} - \hat{A}_{\mathcal{D}}|$$

$$\leq \mathbb{E}_{\mathcal{D},\mathcal{D}'} \sup_f |\hat{A}_{\mathcal{D}'} - \hat{A}_{\mathcal{D}}|$$

$$\leq \mathbb{E}_{\mathcal{D},\mathcal{D}',\sigma} \sup_f |\frac{1}{N} \sum_{i=1}^{N} \sigma_i (-\log f_{\nu(\boldsymbol{x}_i',y_i')}^{y_i'}(\boldsymbol{x}_i') + \log f_{\nu(\boldsymbol{x}_i',y_i')}^{y_i'}(\boldsymbol{0}) + \log f_{\nu(\boldsymbol{x}_i,y_i)}^{y_i}(\boldsymbol{x}_i) - \log f_{\nu(\boldsymbol{x}_i,y_i)}^{y_i}(\boldsymbol{0}))|$$

$$\leq \mathbb{E}_{\mathcal{D},\sigma} \sup_f |\frac{1}{N} \sum_{i=1}^{N} \sigma_i \log f_{\nu(\boldsymbol{x}_i,y_i)}^{y_i}(\boldsymbol{x}_i)| + \mathbb{E}_{\mathcal{D}',\sigma} \sup_f |\frac{1}{N} \sum_{i=1}^{N} \sigma_i \log f_{\nu(\boldsymbol{x}_i',y_i')}^{y_i'}(\boldsymbol{x}_i')|$$

$$+ \mathbb{E}_{\mathcal{D},\sigma} \sup_f |\frac{1}{N} \sum_{i=1}^{N} \sigma_i \log f_{\nu(\boldsymbol{x}_i,y_i)}^{y_i}(\boldsymbol{0})| + \mathbb{E}_{\mathcal{D}',\sigma} \sup_f |\frac{1}{N} \sum_{i=1}^{N} \sigma_i \log f_{\nu(\boldsymbol{x}_i',y_i')}^{y_i'}(\boldsymbol{0})|$$

$$= 2\mathbb{E}_{\mathcal{D},\sigma} \sup_f |\frac{1}{N} \sum_{i=1}^{N} \sigma_i \log f_{\nu(\boldsymbol{x}_i,y_i)}^{y_i}(\boldsymbol{x}_i)| + 2\mathbb{E}_{\mathcal{D},\sigma} \sup_f |\frac{1}{N} \sum_{i=1}^{N} \sigma_i \log f_{\nu(\boldsymbol{x}_i,y_i)}^{y_i}(\boldsymbol{0})|$$

$$\leq 4\mathbb{E}_{\mathcal{D},\sigma} \sup_f |\frac{1}{N} \sum_{i=1}^{N} \sigma_i \log f_{\nu(\boldsymbol{x}_i,y_i)}^{y_i}(\boldsymbol{x}_i)|$$

$$\leq 4 \sum_{s \in \mathcal{S}} \mathbb{E}_{\mathcal{D},\sigma} \sup_f |\frac{1}{N} \sum_{i=1}^{N} \sigma_i \log f_s^{y_i}(\boldsymbol{x}_i)|$$

$$= 4K R_N(G), \tag{23}$$

where $\sigma_i, i = 1, 2, \dots N$ is the Rademacher variable that is uniformly sampled in $\{-1, +1\}$. Combining Eq. (17), Eq. (18), Eq. (21), Eq. (23), $\forall \delta \in (0, 1)$, we have:

$$|(I(X; Y; \nu(X, Y))) - (-\hat{\mathcal{R}}(f^{\dagger}, \nu; \mathcal{D}) + B)| \leq \frac{m}{\sqrt{N}} \sqrt{-2 \log \delta} + 4K \cdot R_N(G) \tag{24}$$

with probability at least $1 - \delta$.

$\square$

### C.3 PROOF OF THM. 3.4

*Proof of Thm. 3.4.*
If $z_s$ can perfectly fit the data distribution of a given subpopulation $s$, *i.e.*, $p(\boldsymbol{x}, y|s) \propto e^{z_s^y(\boldsymbol{x})}$. Let $p(\boldsymbol{x}, y|s) = w \cdot e^{z_s^y(\boldsymbol{x})}, w > 0$. We can get

$$p(y|\boldsymbol{x}; s) = \frac{p(\boldsymbol{x}, y|s)}{\sum_{y'} p(\boldsymbol{x}, y'|s)} = \text{softmax}(z_s^y(\boldsymbol{x})) \tag{25}$$

For the subpopulation-balanced distribution, $p_{bal}(s) = \frac{1}{K}, \forall s \in \mathcal{S}$. We have

$$
\begin{aligned}
p_{bal}(\boldsymbol{x}, y) &= \sum_{s \in \mathcal{S}} p(\boldsymbol{x}, y | s) p_{bal}(s) \\
&= \frac{1}{K} \sum_{s \in \mathcal{S}} p(\boldsymbol{x}, y | s) \\
&= \frac{1}{K} \sum_{s \in \mathcal{S}} w \cdot e^{z_s^y(\boldsymbol{x})} \\
&\propto e^{z^y(\boldsymbol{x})}
\end{aligned}
\tag{26}
$$

And we have

$$
p(y | \boldsymbol{x}) = \frac{p(\boldsymbol{x}, y)}{\sum_{y'} p(\boldsymbol{x}, y')} = \mathrm{softmax}(z^y(\boldsymbol{x}))
\tag{27}
$$

$\square$

### C.4 An Extension of Thm. 3.4

We can extend Thm. 3.4 to make SHE handle an arbitrary specified target distribution over the latent subpopulation $p_{test}(\boldsymbol{x}, y) = \sum_{s \in \mathcal{S}} p_{test}(s) p(\boldsymbol{x}, y | s)$, in the form of a weighted variant of LogSumExp.

**Proposition C.3.** *Supposing that for any subpopulation $s \in \mathcal{S}$, $z_s$ can perfectly fit the data distribution of a given subpopulation $s$, i.e., $p(\boldsymbol{x}, y | s) \propto e^{z_s^y(\boldsymbol{x})}$, then $z = \log \sum_{s \in \mathcal{S}} p_{test}(s) e^{z_s}$ can perfectly fit the subpopulation-balanced overall distribution, i.e., $p_{test}(\boldsymbol{x}, y) \propto e^{z^y(\boldsymbol{x})}$.*

The proof shares the same spirit as the proof of Thm. 3.4.

## D  Additional Discussions of Eq. (2)

Here we would like to explain more about the intuitive understanding of Eq. (2). For ease of reading, here we restate Eq. (2) as follows:

$$
\mathcal{L} = -\frac{1}{N} \sum_{i=1}^{N} \sum_{s \in \mathcal{S}} v_{is} \cdot \log f_s^{y_i}(\boldsymbol{x}_i) - \hat{H}_{\mathcal{D}}(Y | V) - \beta \frac{1}{N} \sum_{i=1}^{N} \mathrm{Div}(\boldsymbol{x}_i).
$$

With respect to $V$, the first term of Eq. (2) increases the weight of the subpopulation that predicts accurately for each sample. The second term makes the classes in each subpopulation as balanced as possible, which prevents the subpopulation from collapsing to the prediction target. And the third term prevents each subpopulation from collapsing to exactly the same prediction and accelerate the optimization. The second and third terms somewhat prevent $V$ from falling into the trivial solutions.

## E  The efficiency and scalability of optimizing $V$

As stated in Sec. 3.4, the subpopulation-weight matrix $V$ has size of $N \times K$. To improve the efficiency of optimizing $V$ and to increase the scalability of our method on large datasets (where $N$ grows larger), we design a batch-specific update approach for $V$. This means that during each iteration, only the elements of $V$ corresponding to the samples in each mini-batch (i.e., $BatchSize \times K$ elements) are updated, while the remaining elements are kept fixed. This approach significantly reduces the number of updated elements in $V$ to $BatchSize \times K$, which is far smaller than the number of parameters in a modern neural network, allowing for scalability even when dealing with large datasets.

# F DETAILED SUPPLEMENT FOR EXPERIMENTS

## F.1 SUPPLEMENTAL DESCRIPTION OF THE EXPERIMENTAL SETUP

### F.1.1 TOY MOTIVATING EXAMPLE (FIG. 2)

In Fig. 2, we visualize a toy motivating example whose prediction goal is to distinguish between circles (semi-transparent) and triangles (non-transparent). For training data, they are sampled from both subpopulation 1 (blue) and subpopulation 2 (red), and the training samples of subpopulation 2 are much less than those of subpopulation 1, *i.e.,*, under subpopulation imbalance. About the test set, it is sampled equally from both subpopulations, *i.e.,*, under subpopulation balance. Specifically, each class in each subpopulation is sampled from the following normal distributions: $N([1, 3], [0.2, 0.2])$ (subpopulation 1, class 1), $N([2, 3], [0.2, 0.2])$ (subpopulation 1, class 2), $N([1.5, 1], [0.2, 0.2])$ (subpopulation 2, class 1), and $N([1.5, 2], [0.2, 0.2])$ (subpopulation 2, class 2), respectively. The sample size of the training set is $200, 200, 10, 10$ in the corresponding order. The sample size of the test set for each normal distribution is $100$. We use a two-layer MLP with 5 hidden neurons as the model for the toy study. The batch size is set to $512$. The toy models are trained using SGD with momentum of 0.9. We train the models for 60 epochs with initial learning rate 0.2.

### F.1.2 TRAINING UNDER SUBPOPULATION IMBALANCE COUPLED WITH CLASS IMBALANCE

For COCO, we conduct training set with both subpopulation imbalance and class imbalance and both balanced test set following the GLT-protocol in Tang et al. (2022). For CIFAR and tiredImageNet, we shuffle the subcategories in the dataset randomly and then sample them in an imbalanced manner, so that they are imbalanced at both the category and subpopulation levels.

### F.1.3 TRAINING UNDER SPURIOUS CORRELATIONS

CelebA (Liu et al., 2015) and Waterbirds (Sagawa et al., 2019) are two datasets that have been widely used to benchmark the robustness of machine learning algorithms to spurious correlations. In the CelebA dataset, there is a high correlation between gender = {male, female} and hair color = {blond, dark}, meaning that the feature gender might be used as a proxy to predict the hair color. In Waterbirds, there is a high correlation between y = {land bird, water bird} and background = {land, water}. The baselines for comparison include GDRO (Sagawa et al., 2019), LfF (Nam et al., 2020), SD (Pezeshki et al., 2021), JTT (Liu et al., 2021), CIM (Taghanaki et al., 2021), and MaskTune (Taghanaki et al., 2022).

We strictly follow the experimental setup in Taghanaki et al. (2022). For the Waterbirds dataset, we train an ImageNet pre-trained ResNet50 with a batch size of 128 for 100 epochs decaying the learning rate by 0.1 after every 30 epochs. For the CelebA dataset, we train an ImageNet pre-trained ResNet50 with a batch size of 512 for 20 epochs with a learning rate of $10^{-4}$. For our SHE , the pre-defined $K$ is set to 2. When the group information is unknown for all methods, we utilize the overall accuracy on a validation set that shares the same distribution as the training set as the selection metric.

### F.1.4 LINEAR PROBING

We train a linear classifier on a frozen pre-trained backbone and measure the quality of the representation through the test accuracy. To eliminate the effect of the skewed distribution in the fine-tuning phase, the classifier is trained on a subpopulation-balanced dataset. Specifically, the performance of the classifier is reported on the basis of pre-trained representations for different amounts of data, including full-shot, 100-shot, and 50-shot. In the fine-tuning phase, we train the linear classifier for 500 epochs with SGD of momentum 0.7 and weight decay 0.0005. The batch size is set to 1000. The learning rate decays exponentially from $10^{-2}$ to $10^{-6}$. The loss function is set to the ordinary cross-entropy loss.

Apple          Banana

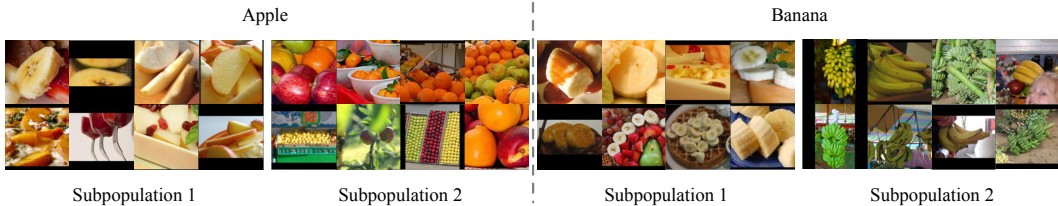

Subpopulation 1     Subpopulation 2     Subpopulation 1     Subpopulation 2

Figure 5: Visualization of learned subpopulations in COCO. To ease the visualization and analysis, we set the number of subpopulations as $K = 2$. Then, after training, we randomly selected 8 images from the two subpopulations we learned about the classes "apple" and "banana" in COCO.

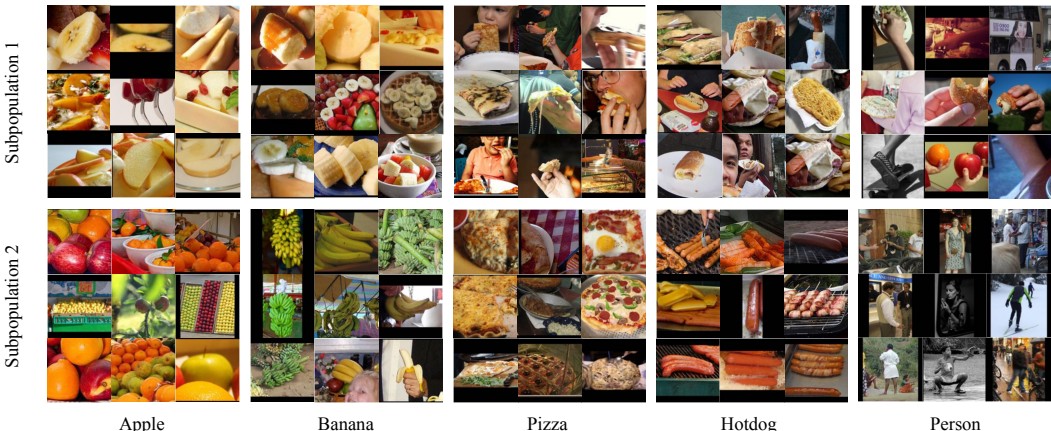

Apple       Banana       Pizza       Hotdog       Person

Figure 6: More visualization of learned subpopulations in COCO. To ease the visualization and analysis, we set the number of subpopulations as $K = 2$. Then, after training, we randomly selected 9 images from the two subpopulations we learned about the classes "apple", "banana", "pizza", "hotdog", and "person" in COCO.

## F.2   MORE VISUALIZATION OF LEARNED SUBPOPULATIONS

### F.2.1   COCO

We present the visualizations of the subpopulations discovered by SHE using the COCO dataset, as depicted in Fig. 5. We also make reference to these visualizations in Sec. 3.3. Here we show more visualizations in Fig. 6. It can be seen that samples from different subpopulations of the same class have obvious semantic differences, further demonstrating that our SHE is able to discover meaningful hidden subpopulation structures in the training data.

### F.2.2   WATERBIRDS

We show some visualizations of the subpopulations learned by SHE on the Waterbirds dataset in Fig. 7. In the Waterbirds dataset, group annotations are constructed based on classes (land bird/water bird) and backgrounds (land/water). The spurious correlation is exhibited by that most of the birds on the land background are land birds and most of the birds on the water background are water birds. If we distinguish subpopulations based on their background, then category imbalance in each subpopulation will still lead the model to learn spurious correlations. Fortunately, however, SHE divide the training data into two meaningful subpopulations: 1) data with spurious correlations and 2) data without spurious correlations, which actually is different from the prior group annotations, as shown in Fig. 7. Such a result may be due to the second term $\hat{H}_{\mathcal{D}}(Y|V)$ of Eq. (2), which increases the entropy of the classes in each subpopulation to make the classes in each subpopulation as balanced

Land Bird          Water Bird          Land Bird          Water Bird

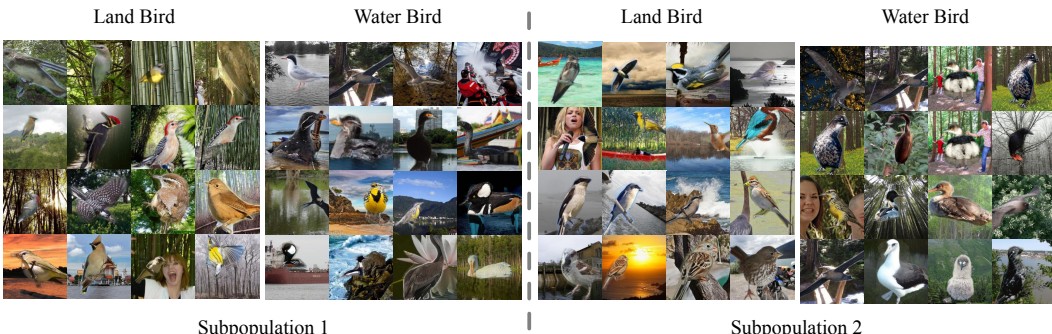

Subpopulation 1                              Subpopulation 2

Figure 7: Visualization of learned subpopulations in Waterbirds. We randomly selected 16 images from the two subpopulations we learned about the classes "land bird", and "water bird" in Waterbirds.

Table 7: Per-split accuracies on COCO (Mean $\pm$ std).

| Method | Many | Medium | Few | Overall |
|---|---|---|---|---|
| ERM | 67.21 $\pm$ 0.24% | 52.22 $\pm$ 0.17% | 37.10 $\pm$ 0.42% | 62.52 $\pm$ 0.32% |
| PaCO | 67.45 $\pm$ 0.31% | 53.33 $\pm$ 0.21% | 36.23 $\pm$ 0.28% | 62.59 $\pm$ 0.24% |
| BCL | 66.89 $\pm$ 0.31% | 53.21 $\pm$ 0.42% | 37.67 $\pm$ 0.24% | 62.83 $\pm$ 0.42% |
| IFL | 67.71 $\pm$ 0.13% | 52.17 $\pm$ 0.24% | 36.82 $\pm$ 0.18% | 62.57 $\pm$ 0.15% |
| DB | 67.35 $\pm$ 0.32% | 52.11 $\pm$ 0.25% | 37.47 $\pm$ 0.36% | 62.72 $\pm$ 0.48% |
| TDE | 66.32 $\pm$ 0.22% | 53.23 $\pm$ 0.28% | 37.02 $\pm$ 0.33% | 62.64 $\pm$ 0.27% |
| ETF-DR | **67.93 $\pm$ 0.17%** | 51.34 $\pm$ 0.22% | 37.59 $\pm$ 0.12% | 62.45 $\pm$ 0.37% |
| LfF | 66.74 $\pm$ 0.34% | 52.34 $\pm$ 0.26% | 36.01 $\pm$ 0.38% | 62.06 $\pm$ 0.83% |
| Focal | 66.23 $\pm$ 0.42% | 52.41 $\pm$ 0.28% | 35.79 $\pm$ 0.36% | 61.67 $\pm$ 0.53% |
| EIIL | 66.87 $\pm$ 0.18% | 52.79 $\pm$ 0.32% | 37.06 $\pm$ 0.28% | 62.61 $\pm$ 0.33% |
| ARL | 67.32 $\pm$ 0.25% | 53.34 $\pm$ 0.17% | 37.18 $\pm$ 0.24% | 62.48 $\pm$ 0.22% |
| GRASP | 67.13 $\pm$ 0.11% | 53.26 $\pm$ 0.16% | 37.29 $\pm$ 0.32% | 62.73 $\pm$ 0.25% |
| JTT | 66.93 $\pm$ 0.26% | 51.24 $\pm$ 0.35% | 36.48 $\pm$ 0.27% | 62.32 $\pm$ 0.75% |
| MaskTune | 64.48 $\pm$ 0.31% | 50.11 $\pm$ 0.35% | 33.27 $\pm$ 0.19% | 60.23 $\pm$ 0.73% |
| SHE | 67.71 $\pm$ 0.32% | **53.50 $\pm$ 0.26%** | **42.09 $\pm$ 0.28%** | **64.56 $\pm$ 0.24%** |

as possible. Such two subpopulations plus further subpopulation rebalancing can effectively prevent the model from relying on spuriously correlated features for prediction.

### F.3    COMPLETE RESULTS OF TAB. 3

We provide the complete experimental results (Mean $\pm$ Std) of all baselines of Tab. 3 in Tab. 7.

### F.4    COMPLETE RESULS OF TAB. 6

We provide the complete experimental results (Mean $\pm$ Std) of Tab. 6 in Tab. 8.

### F.5    MORE RESULTS ON VARYING THE LATENT SUBPOPULATION NUMBER $K$

In Fig. 4(a), we show ablation study on $K$ on COCO. Here we give the complete experimental results on more dataset, *i.e.*, CIFAR-IR100, CIFAR-IR50, CIFAR-IR20, and tiredImageNet, in Fig. 8. When $K = 1$, Eq. (2) degenerates to the cross-entropy loss, and our SHE degenerates to the ERM performance. When $K > 1$, SHE has a significant improvement over ERM and shows some robustness to the value of $K$ in general.

Table 8: LoRA fine-tuning under pre-trained models on COCO (Mean ± std).

| Method | CLIP | ALIGN | AltCLIP |
|---|---|---|---|
| Zero-shot | 76.59 ± 0.00% | 78.45 ± 0.00% | 82.55 ± 0.00% |
| ERM | 84.32 ± 0.14% | 83.38 ± 0.12% | 84.85 ± 0.07% |
| PaCO | 84.38 ± 0.11% | 83.54 ± 0.15% | 85.06 ± 0.16% |
| BCL | 84.48 ± 0.06% | 83.32 ± 0.11% | 85.06 ± 0.05% |
| IFL | 84.55 ± 0.06% | 83.40 ± 0.05% | 84.83 ± 0.06% |
| DB | 84.46 ± 0.11% | 83.14 ± 0.13% | 84.14 ± 0.40% |
| TDE | 84.37 ± 0.10% | 83.42 ± 0.05% | 84.51 ± 0.19% |
| ETF-DR | 84.26 ± 0.15% | 83.36 ± 0.09% | 84.78 ± 0.05% |
| LfF | 84.17 ± 0.05% | 83.12 ± 0.07% | 84.20 ± 0.02% |
| Focal | 83.87 ± 0.10% | 82.77 ± 0.12% | 83.84 ± 0.14% |
| EIIL | 84.23 ± 0.12% | 83.21 ± 0.07% | 84.34 ± 0.05% |
| ARL | 84.02 ± 0.08% | 82.97 ± 0.17% | 84.07 ± 0.07% |
| GRASP | 84.32 ± 0.16% | 83.14 ± 0.04% | 84.44 ± 0.13% |
| JTT | 84.45 ± 0.09% | 83.09 ± 0.02% | 84.40 ± 0.15% |
| MaskTune | 83.27 ± 0.10% | 82.54 ± 0.13% | 83.52 ± 0.20% |
| **SHE** | **85.39 ± 0.06%** | **84.23 ± 0.06%** | **85.69 ± 0.11%** |

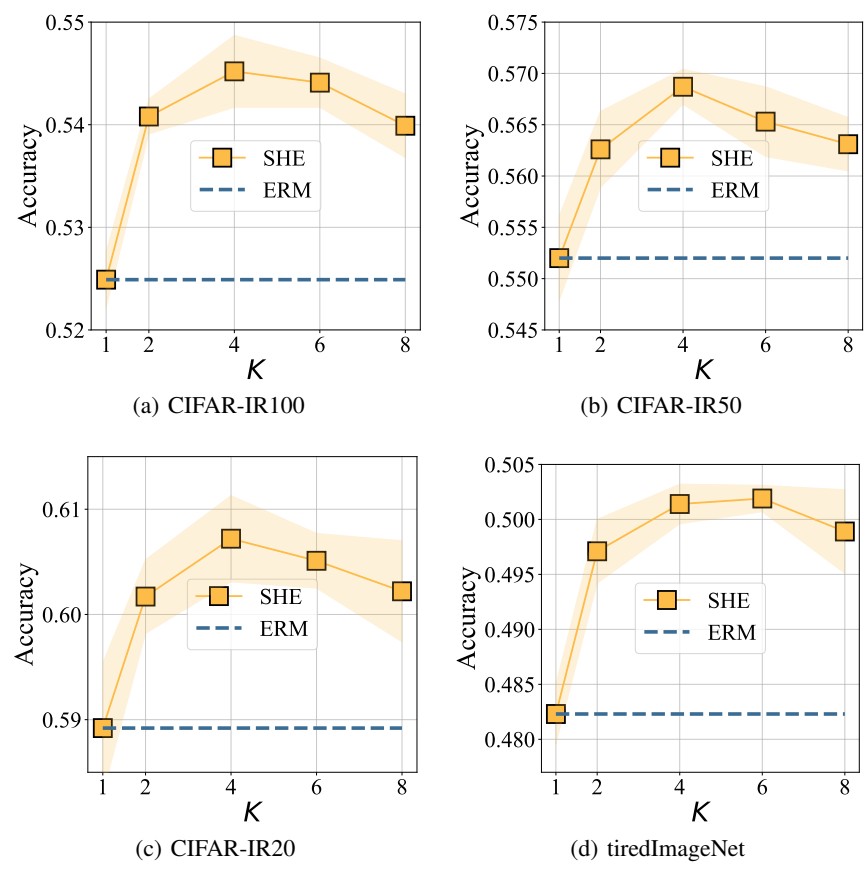

(a) CIFAR-IR100  (b) CIFAR-IR50

(c) CIFAR-IR20  (d) tiredImageNet

Figure 8: Performance of SHE and ERM with varying subpopulation number $K$ on CIFAR-IR100, CIFAR-IR50, CIFAR-IR20, and tiredImageNet.

Table 9: A series of baselines by combining different strategy permutations

| Method abbreviation | Clustering | | | Rebalancing | | Training strategy | |
|---|---|---|---|---|---|---|---|
| | K-means | OT | Prediction | Resample | Reweight | Simultaneous | Sequential |
| K-1 | ✓ | | | ✓ | | ✓ | |
| K-2 | ✓ | | | ✓ | | | ✓ |
| K-3 | ✓ | | | | ✓ | ✓ | |
| K-4 | ✓ | | | | ✓ | | ✓ |
| O-1 | | ✓ | | ✓ | | ✓ | |
| O-2 | | ✓ | | ✓ | | | ✓ |
| O-3 | | ✓ | | | ✓ | ✓ | |
| O-4 | | ✓ | | | ✓ | | ✓ |
| P-1 | | | ✓ | ✓ | | ✓ | |
| P-2 | | | ✓ | ✓ | | | ✓ |
| P-3 | | | ✓ | | ✓ | ✓ | |
| P-4 | | | ✓ | | ✓ | | ✓ |

Table 10: Comparison with more clustering and rebalancing components. Bold indicates superior results. The meaning of some method abbreviations can be found in Tab. 9.

| Method | COCO | CIFAR-IR100 | CIFAR-IR50 | CIFAR-IR20 |
|---|---|---|---|---|
| ERM | $62.52 \pm 0.32\%$ | $52.49 \pm 0.27\%$ | $55.20 \pm 0.41\%$ | $58.92 \pm 0.62\%$ |
| K-1 | $61.59 \pm 0.41\%$ | $51.43 \pm 0.34\%$ | $54.56 \pm 0.47\%$ | $57.97 \pm 0.35\%$ |
| K-2 | $62.63 \pm 0.35\%$ | $52.57 \pm 0.40\%$ | $55.03 \pm 0.18\%$ | $59.03 \pm 0.28\%$ |
| K-3 | $61.77 \pm 0.57\%$ | $51.71 \pm 0.26\%$ | $54.29 \pm 0.37\%$ | $58.21 \pm 0.32\%$ |
| K-4 | $62.48 \pm 0.27\%$ | $52.52 \pm 0.34\%$ | $55.27 \pm 0.27\%$ | $58.81 \pm 0.27\%$ |
| O-1 | $62.56 \pm 0.39\%$ | $52.52 \pm 0.44\%$ | $55.13 \pm 0.42\%$ | $58.85 \pm 0.20\%$ |
| O-2 | $62.49 \pm 0.21\%$ | $52.44 \pm 0.42\%$ | $55.22 \pm 0.21\%$ | $58.85 \pm 0.49\%$ |
| O-3 | $62.54 \pm 0.32\%$ | $52.47 \pm 0.35\%$ | $55.15 \pm 0.38\%$ | $58.90 \pm 0.51\%$ |
| O-4 | $62.58 \pm 0.36\%$ | $52.43 \pm 0.30\%$ | $55.08 \pm 0.58\%$ | $58.99 \pm 0.44\%$ |
| P-1 | $61.32 \pm 0.28\%$ | $51.27 \pm 0.37\%$ | $54.31 \pm 0.42\%$ | $57.62 \pm 0.31\%$ |
| P-2 | $62.47 \pm 0.46\%$ | $52.25 \pm 0.43\%$ | $54.96 \pm 0.49\%$ | $58.71 \pm 0.26\%$ |
| P-3 | $61.58 \pm 0.14\%$ | $51.53 \pm 0.29\%$ | $54.00 \pm 0.18\%$ | $58.13 \pm 0.55\%$ |
| P-4 | $62.30 \pm 0.38\%$ | $52.47 \pm 0.33\%$ | $55.25 \pm 0.26\%$ | $58.57 \pm 0.33\%$ |
| SHE-RS | $64.03 \pm 0.27\%$ | $53.97 \pm 0.38\%$ | $56.17 \pm 0.50\%$ | $60.25 \pm 0.54\%$ |
| SHE-RW | $63.82 \pm 0.22\%$ | $53.90 \pm 0.41\%$ | $56.05 \pm 0.27\%$ | $60.19 \pm 0.27\%$ |
| **SHE** | $\mathbf{64.56 \pm 0.24\%}$ | $\mathbf{54.52 \pm 0.35\%}$ | $\mathbf{56.87 \pm 0.17\%}$ | $\mathbf{60.72 \pm 0.41\%}$ |

## F.6 COMPARISON WITH MORE CLUSTERING AND REBALANCING COMPONENTS

To further demonstrate the superiority of SHE, we compare it with a series of clustering and rebalancing strategies. The clustering strategies include K-means, optimal transport clustering, and direct using the ERM predictions. The rebalancing strategies include reweighting and resampling. And the training strategies include: 1) the simultaneous strategy: at each epoch of training, the clustering results and rebalancing weights are updated; 2) the sequential strategy: first train an ERM model and obtain clustering results, then conduct rebalancing to retrain a model based on the clustering results. We construct a series of baselines by combining these strategy permutations, which are presented in Tab. 9. To confirm the effectiveness of the subpopulation-balanced prediction by Thm. 3.4, we also construct two variants of SHE: SHE-RS (dynamic resampling based on the learned subpopulations during training), and SHE-RS (dynamic reweighting based on the learned subpopulations during training). We show the performance of all these baselines and SHE on COCO, CIFAR-R100, CIFAR-R50, and CIFAR-R20 in Tab. 10. Our method still achieves the best results very clearly, demonstrating the superiority of our method for subpopulation discovery and rebalancing.

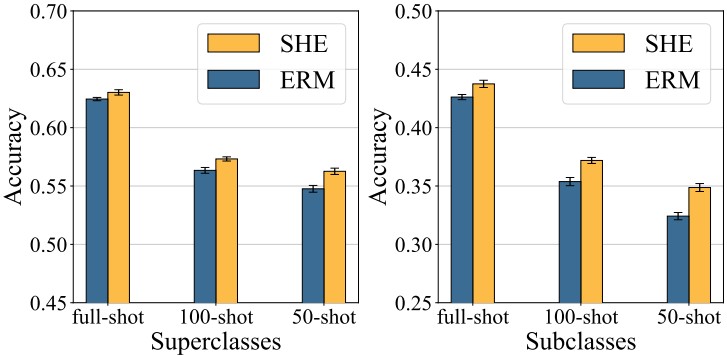

Figure 9: Linear probing performance of SHE and ERM on CIAFR-IR100 superclasses (left) and subclasses (right) under different shot settings.

Table 11: Worst-case performance on CIFAR-IR100/50/20.

| Method | ERM | PaCO | BCL | IFL | DB | TDE | ETF-DR | LfF | Focal | JTT | MT | **SHE** |
|---|---|---|---|---|---|---|---|---|---|---|---|---|
| CIFAR-IR100 | 22.34% | 23.49% | 23.83% | 22.17% | 22.87% | 22.53% | 22.96% | 20.74% | 19.45% | 21.38% | 19.78% | **27.48%** |
| CIFAR-IR50 | 26.47% | 27.17% | 27.58% | 26.71% | 26.93% | 26.68% | 27.43% | 23.67% | 23.16% | 24.58% | 23.17% | **31.24%** |
| CIFAR-IR20 | 37.10% | 38.14% | 38.07% | 27.45% | 37.77% | 37.29% | 37.93% | 34.73% | 34.25% | 35.28% | 34.64% | **41.19%** |

Table 12: Compared with other alternative ways of doing inference.

| Method | COCO | CIFAR-IR100 | CIFAR-IR50 | CIFAR-IR20 |
|---|---|---|---|---|
| ERM | $62.52 \pm 0.32\%$ | $52.49 \pm 0.27\%$ | $55.20 \pm 0.41\%$ | $58.92 \pm 0.62\%$ |
| $SHE_{EIIL}$ | $62.82 \pm 0.27\%$ | $52.63 \pm 0.22\%$ | $55.36 \pm 0.37\%$ | $59.21 \pm 0.48\%$ |
| $SHE_{SimAvg}$ | $64.54 \pm 0.17\%$ | $54.44 \pm 0.41\%$ | $56.78 \pm 0.28\%$ | $60.66 \pm 0.32\%$ |
| SHE | $\mathbf{64.56 \pm 0.24\%}$ | $\mathbf{54.52 \pm 0.35\%}$ | $\mathbf{56.87 \pm 0.17\%}$ | $\mathbf{60.72 \pm 0.41\%}$ |

### F.7 LINEAR PROBING PERFORMANCE

To quantitatively evaluate the representation quantity of different methods, we conduct linear probing experiments on CIFAR-IR100 following the literature of self-supervised learning (Chen et al., 2020; He et al., 2020). To eliminate the subpopulation imbalance effect, the linear classifier is trained on a balanced dataset on top of the fixed feature extractor. In Fig. 9, we show the linear probing performance of both superclasses and subclasses on CIFAR-IR100 under different shots. As can be seen, our SHE consistently exceeds the ERM baseline for all settings, especially for the linear probing performance of fine-grained classes with the improvement of 1.13%, 1.81%, and 2.45% on full-shot, 100-shot, and 50-shot. This indicates that our SHE actually captures better and more generalized representations.

### F.8 WORST CASE PERFORMANCE.

To further validate the efficacy of our SHE in enhancing the learning capability of rare samples, we present the worst case performance on CIFAR-IR100, CIFAR-IR50, and CIFAR-IR20 in Tab. 11. It is evident that our method has achieved significantly superior results compared to other baselines, further substantiating its remarkable effectiveness in mitigating the issue of subpopulation imbalance.

### F.9 MORE EXPLORATION ON ALTERNATIVE WAYS OF DOING INFERENCE

Regarding alternative ways of doing inference, we validate some simple variants like applying group-invariant learning on the learned subpopulation (marked as '$SHE_{EIIL}$') or just averaging the logits across $f_s$ (marked as '$SHE_{SimAvg}$'). As shown in Tab. 12, SHE significantly outperforms $SHE_{EIIL}$, while achieving comparable results with $SHE_{SimAvg}$. This is because, applying the Normalized

Table 13: Compatision between w/ or w/o LA on COCO where both class imbalance and subpopulation imbalance co-exist.

| Method | Acc |
|--------|-----|
| ERM | 63.57 ± 0.34% |
| SHE | **65.13 ± 0.22 %** |
| LA | 66.47 ± 0.27% |
| SHE$_{w/ LA}$ | **68.11 ± 0.27%** |

Table 14: Compared with other alternative ways of optimizing $V$.

| Method | COCO | CIFAR-IR100 | CIFAR-IR50 | CIFAR-IR20 |
|--------|------|-------------|------------|------------|
| ERM | 62.52 ± 0.32% | 52.49 ± 0.27% | 55.20 ± 0.41% | 58.92 ± 0.62% |
| SHE$_{model based V}$ | 63.22 ± 0.13% | 53.28 ± 0.36% | 55.81 ± 0.17% | 59.72 ± 0.38% |
| SHE$_{EM}$ | 64.52 ± 0.31% | 54.47 ± 0.26% | **56.88 ± 0.22%** | 60.65 ± 0.33% |
| SHE | **64.56 ± 0.24%** | **54.52 ± 0.35%** | 56.87 ± 0.17% | **60.72 ± 0.41%** |

Table 15: Performance on datasets for spurious correlations.

| Method | Group Info (Train / Val) | CelebA | Waterbirds |
|--------|--------------------------|--------|------------|
| GDRO | Yes / Yes | 88.3% / 91.8% | 91.4% / 93.5% |
| SHE$_{w/goldlabels}$ | Yes / Yes | **88.4%** / 91.3% | **91.6%** / 93.2% |
| ERM | No / No | 47.2% / 95.6% | 74.9% / 98.1% |
| SHE | No / No | 77.7% / 92.0% | **82.0%** / 91.3% |
| SHE$_{w/GDRO}$ | No / No | **77.9%** / 91.7% | 81.9%/ 91.3% |

Weighted Geometric Mean (NWGM) approximation (Baldi & Sadowski, 2013; Xu et al., 2015), LogSumExp can be approximated as simple summation, which is equivalent to simple averaging the logits. Here, we use LogSumExp because of its advantage of being more theoretically rigorous with its numerical stability.

### F.10 MORE EXPLORATION ON THE OPTIMIZATION APPROACH FOR $V$

In terms of alternative approaches for optimizing $V$, we examined different variations of SHE. Firstly, we utilized a 2-layer MLP to learn $V$ from image features, referred to as SHEmodel based $V$. Secondly, we employed an EM-style approach to alternately learn $V$ and $f_s$, referred to as SHEEM. As indicated in Table 14, SHEmodel based $V$ exhibits a noticeable performance degradation compared to SHE. This can be attributed to the fact that $\nu$ in Definition 3.1 is dependent on both the input $x$ and the label $y$, whereas SHEmodel based $V$ can only learn the data partition from $x$. On the other hand, SHE$_{EM}$ demonstrates comparable results with SHE, yet SHE is simpler and superior, thus confirming the effectiveness of the proposed optimization approach.

Since SHE$_{model based V}$, which learns $V$ solely from $X$, does not satisfy our formulation, we similarly construct SHE$_{model based V from X,Y}$ to learn $V$ from both $X$ and $Y$. The experimental results are presented in Tab. 16. SHE$_{model based V from X,Y}$ outperforms the variant that solely learn $V$ from $X$, as it aligns better with our formulation. However, due to the batch-wise updates of the network learning $V$ under both variants, it is challenging to accurately compute the subpopulation allocation for samples not in the batch, yielding the inaccuracy when calculating the entropy term in Eq. (2) and thus the performance degradation compared to SHE.

Table 16: Performance of two variants of model-based methods to learn $V$.

| Method | CIFAR-IR100 | CIFAR-IR50 | CIFAR-IR20 |
|---|---|---|---|
| ERM | 52.49% | 55.20% | 58.92% |
| SHE$_{\text{model based V from X}}$ | 53.28% | 55.81% | 59.72% |
| SHE$_{\text{model based V from X,Y}}$ | 54.08% | 56.23% | 60.14% |
| SHE | 54.52% | 56.87% | 60.72% |

### F.11 MORE ANALYSIS ON RICHER IMBALANCE CONTEXTS

SHE is primarily designed to address subpopulation imbalance issues and optimize overall performance under subpopulation balanced distributions. However, it does not specifically focus on class imbalance and worst-case performance. Therefore, we combined SHE with LA or GDRO in the presence of class imbalance and spurious correlation, as discussed in Sec. 4.3. In this context, we present additional results of using SHE alone when dealing with class imbalance and spurious correlation settings in Tab. 13 and Tab. 15, respectively.

From Tab. 13, it is evident that our SHE achieves significant improvements whether applied on top of ERM or LA. The results in Tab. 15 demonstrate that SHE achieves comparable performance to SHEw/ GDRO. We report the results of SHEw/ GDRO in Sec. 4.3 to maintain consistency with the objective of the spurious correlation task.

### F.12 COMPUTATIONAL COST

Considering the computational cost, SHE (according to Eq. (2)) and Appx. E)incurs additional computational overhead compared to ERM due to: 1) Weighted summation of the cross-entropy loss, 2) Computation of empirical entropy and the regularization term, and 3) Updating a matrix of size $BatchSize \times K$. The computational expenses for these three components are all far smaller than the training cost of a modern deep neural network. Tab. 17 presents a comparison of training duration for different methods.

Table 17: Training time of 200 epochs on CIFAR100-IR100.

| Method | Training time (Minutes) | $\Delta$ ERM |
|---|---|---|
| ERM | 89 | - |
| PaCo | 104 | +16.8% |
| BCL | 110 | +23.6% |
| DB | 102 | +14.6% |
| TDE | 95 | +6.7% |
| IFL | 142 | +60% |
| JTT | 101 | +13.5% |
| LfF | 100 | +12.4% |
| MaskTune | 105 | +17.9% |
| SHE | 98 | +11.1% |

### F.13 BALANCED-CASE PERFORMANCE

We provide the result of SHE and ERM under the subpopulation balanced senario (both train and test) in Tab. 18. Our SHE get slightly better performance even in the balanced case.

Table 18: Balanced case performance.

| Method | CIFAR | tieredImageNet |
|---|---|---|
| ERM | 74.32% | 68.26% |
| SHE | 74.75% | 68.83% |

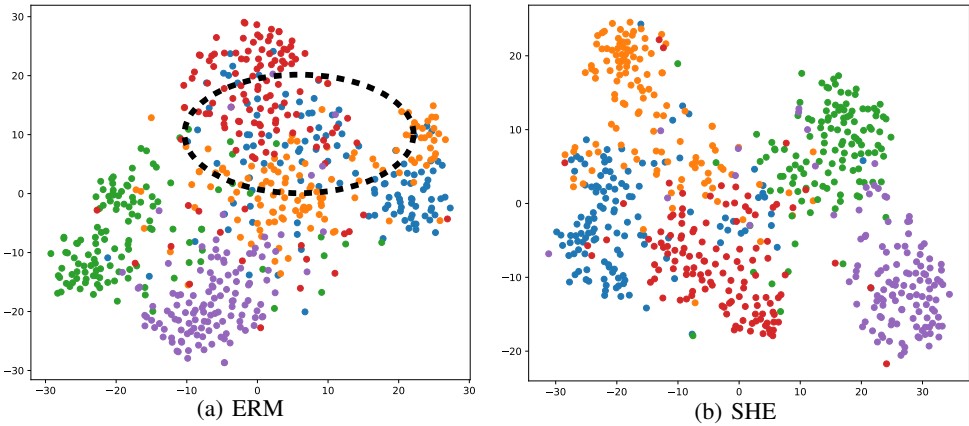

Figure 10: T-SNE visualization of 5 subpopulations whin the same class on CIFAR100. Models are trained on CIFAR-IR20.

### F.14 IN-DISTRIBUTION PERFORMANCE

We present results in Tab. 19 under the scenario where the test set shares the same distribution as the training set, indicating that SHE achieves comparable or slightly better results than ERM in this situation.

Table 19: Performance when the test set shares the same distribution as the training set.

| Method | CIFAR-IR100 | CIFAR-IR50 | CIFAR-IR20 |
|--------|-------------|------------|------------|
| ERM    | 71.21%      | 70.51%     | 68.94%     |
| SHE    | 71.07%      | 70.72%     | 69.25%     |

### F.15 REVERSE-DISTRIBUTION PERFORMANCE

Although Prop. C.3 tells the direction of handling an arbitrary specified target distribution. a challenge in application lies in correctly aligning the learned subpopulations with their corresponding test distributions. To validate the effectiveness of Proposition B.3, we construct an experimental scenario where the test distribution is the reverse imbalanced distribution of the training set. Therefore, we can correspondingly reverse-sort the learned subpopulations by sample size to align with the test distribution. Tab. 20 illustrates the results.

Table 20: Reverse-Distribution Performance.

| Method | CIFAR-IR100 | CIFAR-IR50 | CIFAR-IR20 |
|--------|-------------|------------|------------|
| ERM    | 38.37%      | 42.96%     | 50.70%     |
| SHE    | 43.53%      | 46.74%     | 53.82%     |
| SHE$_{\text{w/ test distribution}}$ | 45.64% | 48.02% | 54.64% |

### F.16 T-SNE FEATURE VISUALIZATION OF SUBPOPULATIONS

In Fig. 10, we illustrate the t-SNE features of five subpopulations within the same category on CIFAR100. This demonstrates that SHE is effective in preventing minority subpopulations(orange and blue) from being overshadowed by others.

## G  LIMITATIONS AND FUTURE EXPLORATIONS

Similar to some clustering methods (*e.g.*, K-means), our method relies on a predefined number of subpopulations $K$. Choosing an inappropriate value of $K$ may lead to overfitting or underfitting of the subpopulation structure and affect the performance and interpretability of the method. We study the scenario in this paper where all subpopulation annotations are invisible. When part of the subpopulation annotations are visible, how to further improve the performance and robustness of machine learning algorithms by leveraging partially labeled data to refine the subpopulation structure needs to be further explored.

