# OpenReview forum: "On Harmonizing Implicit Subpopulations"
_ICLR.cc/2024/Conference — ICLR 2024 poster_

### Official Review · Reviewer_Dvju · 2023-10-27

**Soundness:** 3 good
**Presentation:** 2 fair
**Contribution:** 3 good
**Rating:** 6
**Confidence:** 3

**Summary:**

This paper proposes an approach (SHE) to learning a classifier that performs well for a set of latent subpopulations that are not uniformly represented in the training data. The approach is to learn subpopulation structure through an “optimal data partition” with maximal conditional entropy of the label given the learned subpopulations, effectively identifying subpopulations with balanced label distributions. Evaluation is conducted with respect to a balanced marginal distribution over the subpopulations. An extensive empirical evaluation is conducted using the COCO, CIFAR-100, and ImageNet datasets. The experiments involve comparisons to alternative approaches and ablation studies.

**Strengths:**

* The presentation of the toy example in Figure 2 is clear and compelling for motivating the problem.
* The theoretical analysis in section 3 is sound (to the best of my knowledge) and the method straightforward to implement.
* The empirical evaluation is extensive, with comparisons to several baseline approaches in several settings, including on computer vision datasets that have been variably rebalanced, and in the context of fine-tuning large multimodal foundation models.

**Weaknesses:**

* The paper is challenging to read, and borders on unreadable at times, due to numerous grammatical errors and unusual choices of terminology, particularly in the abstract and the first two sections. As a reader, I was not able to understand the problem that this paper aims to solve until reviewing the mathematical presentation in section 3. While I like this paper overall, this is enough for me to argue that this paper should not be published without substantial revision.
* It is not clear how model complexity and finite-sample considerations interact with the core arguments of the work. For example, the toy example in Figure 2 relies on the use of a linear model for the ERM and subpopulation-specific models. However, a more complex, non-linear ERM model could still, in-principle, fit the data well, even if it might not due to data insufficiency for the underrepresented subpopulations. It’s not obvious how this consideration surfaces in the theoretical presentation nor in the experiments.

**Questions:**

* How does SHE compare to baselines in-domain, i.e., in cases when both the training and testing data are imbalanced? If I understand correctly, the claim of Proposition 3.2 is that SHE should not underperform ERM (in the limit) but it seems that this is not evaluated in the experiments because of the focus on balanced test distributions.
* SHE is motivated to address a particular distribution shift problem where there is subpopulation shift over a set of subpopulations with balanced label distributions. How would SHE perform under other notions of subpopulation shift, where it is not assumed that the label distributions are balanced within subpopulations?
* Could SHE be extended to handle an arbitrary specified target distribution over the latent subpopulation? Would this be as simple as weighting the subpopulation components of the LogSumExp operation?
* Is there a reason why the model-based $V$ variant of SHE could not depend on both $X$ and $Y$? My interpretation of the method was that $V$ is not used at test time, so this might allow for a scalable model-based subpopulation mapping that matches the performance of the full $N$x$K$ matrix.

---

> ### Author Response · Authors · 2023-11-16
> **Response to Reviewer Dvju [1/3]**
>
> > W1: The paper is challenging to read, and borders on unreadable at times, due to numerous grammatical errors and unusual choices of terminology, particularly in the abstract and the first two sections. As a reader, I was not able to understand the problem that this paper aims to solve until reviewing the mathematical presentation in section 3. While I like this paper overall, this is enough for me to argue that this paper should not be published without substantial revision.
>
> **A:** Following the reviewer's suggestions and other reviewers' advice on writting, we have comprehensively revised our submission to enhance clarity and readability. We summarize the main improvement as below:
> - Checking and correcting some grammar errors, replacing some potentially obscure terminologies, such as "unperceived" "non-causal factors from the causal prediction" and "unitary", with more accessible alternatives. We also avoid mixing certain synonymss, such as "unknown" and "implicit".
> - Refining the first paragraph to focus on introducing and emphasizing the problem we aim to solve.
> - Adjusting symbols in Table 1.
> - Fine-tuning and reinforcing the exposition of our intuition in Section 1 and Section 3.
> - Explicitly stating potential underlying assumptions in the problem formulation and the theory part.
> - Formally defining and supplementing introductions to some cited definitions and mathematical tools.
>
> It is welcome and appreciated **if you have more suggestions or if you can specify the concrete parts about the writing that you believe require modification for improvement**? Your detailed feedback will be highly valuable in ensuring the clarity and quality of the paper.
>
> > W2: It is not clear how model complexity and finite-sample considerations interact with the core arguments of the work. For example, the toy example in Figure 2 relies on the use of a linear model for the ERM and subpopulation-specific models. However, a more complex, non-linear ERM model could still, in-principle, fit the data well, even if it might not due to data insufficiency for the underrepresented subpopulations. It’s not obvious how this consideration surfaces in the theoretical presentation nor in the experiments.
>
> **A:** We would like to kindly clarify:
>
> - In Theorem 3.3, our bound incorporates both the **sample size $N$** and the **Rademacher complexity $R_N(G)$**. **As the sample size increases, the bound becomes tighter**. Regarding the model complexity, a lower model complexity generally leads to a lower Rademacher complexity, which results in a tighter bound. However, reducing complexity weakens the model's ability to fit the training data, thereby affecting the optimization of the empirical risk. Therefore, when choosing the model complexity, there exists a **tradeoff** between **a tight bound and effective risk optimization**.
>
> - The model used in Figure 2 is **not a linear model but rather a nonlinear 2-layer MLP** (introduced in Appendix E), and the auxiliary dashed lines in the figure are intended to provide a rough illustration of the model's decision boundaries. And all experiments in Section 4 and the appendix are **based on modern deep networks**, showcasing the consistent gain of SHE compared to other baselines under the same complex model architectures.
>
> - Empirical findings **on complex deep models** highlight the superiority of our method, **aligning with the observations depicted in Figure 2**.
>     - Extensive empirical evidence underscores the effectiveness of our method in subpopulation discovery, as evidenced by **NMI scores in Figure 3 and Figure 4(d)**, along with the **visualization of learned subpopulations in Figure 5-7**.
>
>     - We have incorporated **T-SNE visualizations** for different subpopulations within the same category of CIFAR100 in **Figure 10** of the revised version, indicating that our method is effective in preventing the submergence of minority subpopulations compared to ERM.

---

> ### Author Response · Authors · 2023-11-16
> **Response to Reviewer Dvju [2/3]**
>
> > Q1: How does SHE compare to baselines in-domain, i.e., in cases when both the training and testing data are imbalanced? If I understand correctly, the claim of Proposition 3.2 is that SHE should not underperform ERM (in the limit) but it seems that this is not evaluated in the experiments because of the focus on balanced test distributions.
>
> **A:** We present results in the table below under the scenario **where the test set shares the same imabalanced distribution as the training set**. The results indicate that SHE can achieve **comparable or slightly better** results than ERM in this situation. We have also included this experiment in **Table 19** of the revised version. Note that, we should clarify that such an evaluation is actually beyond the focus of our study.
>
> | Method | CIFAR-IR100 | CIFAR-IR50 | CIFAR-IR20 |
> | :--------: | :--------: | :--------: |:--------: |
> | ERM     | 71.21%     | 70.51%     | 68.94%     |
> | SHE     | 71.07%     | 70.72%     | 69.25%     |
>
> In order to **further investigate the empirical evidence supporting Proposition 3.2**, we try to mitigate the impact of the LogSumExp operation. This is because the **LogSumExp operation (Theorem 3.4)** aims to fit a balanced distribution and may introduce a certain **shift at the classifier level** (in cases of test set imbalance). Hence, we conduct **linear probing experiments (Figure 9)** to compare representation quality, aiming to remove the impact of the LogSumExp operation. The results, as shown in the tables below, indicate that SHE exhibits significantly improved representations compared to ERM.
>
>
> [**Linear probing performance of subclasses (training on CIFAR-IR100)**]
> | Method | 50-shot |100-shot | Full-shot|
> | :--------: | :--------: | :--------: |:--------: |
> | ERM     |32.42%     | 35.38%     | 42.62%     |
> | SHE     | 34.87%     |37.19%     |  43.75%    |
>
> [**Linear probing performance of superclasses (training on CIFAR-IR100)**]
> | Method | 50-shot |100-shot | Full-shot|
> | :--------: | :--------: | :--------: |:--------: |
> | ERM     | 54.76%     |56.34%     | 62.45%    |
> | SHE     | 56.26%     |57.32%     | 63.02%    |
>
> We also present results under **balanced training data** (**Table 18** of the revised version), ensuring both the training and test sets follow the same distribution and adhere to the LogSumExp operation, which further confirming Proposition 3.2.
>
>
> [**Balanced case performance**]
> | Method | CIFAR | tieredImageNet|
> | -------- | -------- | -------- |
> | ERM     | 74.32% | 68.26%|
> | SHE     | 74.75% | 68.83%|
>
> > Q2: SHE is motivated to address a particular distribution shift problem where there is subpopulation shift over a set of subpopulations with balanced label distributions. How would SHE perform under other notions of subpopulation shift, where it is not assumed that the label distributions are balanced within subpopulations?
>
> **A:** We would like to kindly clarify that in **Table 4 (left) and Table 13**, we actually have included experiments under scenarios where **both subpopulations and classes are imbalanced**. The consistent average improvement of over 1.5% than the top-performing baseline demonstrate the effectiveness of our method under such a kind of coupled distribution shift.
>
> > Q3: Could SHE be extended to handle an arbitrary specified target distribution over the latent subpopulation? Would this be as simple as weighting the subpopulation components of the LogSumExp operation?
>
> **A:** Yes, we can extend Theorem 3.4 to make SHE handle an arbitrary specified target distribution over the latent subpopulation, in the form of **a weighted variant of LogSumExp**, i.e, $\log\sum_{s\in\mathcal{S}}p_{test}(s)e^{z_s}$. We provide this extension in the revised version as **Proposition C.3**.
>
> However, a **challenge** in application lies in correctly **aligning the learned subpopulations with their corresponding test distributions**. To validate the effectiveness of Proposition C.3, we conduct an experimental scenario where the test distribution is the **reverse imbalanced distribution** of the training set. Therefore, we can correspondingly reverse-sort the learned subpopulations by sample size to align with the test distribution. The table below illustrates the results, which also have been included in **Table 20** of the revised version.
>
> | Method | CIFAR-IR100 | CIFAR-IR50 | CIFAR-IR20 |
> | :--------: | :--------: | :--------: |:--------: |
> | ERM     | 38.37%     | 42.96%     | 50.70%     |
> | SHE     | 43.53%     | 46.74%     | 53.82%     |
> | SHE$_\mathrm{w/\ test\ distribution}$     | 45.64%     | 48.02%     | 54.64%     |

---

> ### Author Response · Authors · 2023-11-16
> **Response to Reviewer Dvju [3/3]**
>
> > Q4: Is there a reason why the model-based variant of SHE could not depend on both $X$ and $Y$? My interpretation of the method was that  is not used at test time, so this might allow for a scalable model-based subpopulation mapping that matches the performance of the full $N\times K$ matrix.
>
> **A:** In the 4th paragraph of Section 4.4 and Figure 4\(c) of our initial submission, we designed the model-based variant of SHE to only utilize $X$ as input to learn $V$. We have added clarification about this in the revised version to avoid misunderstanding. Additionally, it can depend on both $X$ and $Y$. To fully answer the reviewer's question, here we introduce another **model-based variant of SHE that takes both $X$ and $Y$ as inputs**. The experimental results (**Table 16** of the revised version) are presented in the table below. SHE$_\mathrm{model\ based\ V\ from\ X,Y}$ outperforms the variant that solely learn $V$ from $X$, as it aligns better with our formulation. However, due to the **batch-wise updates** of the network learning $V$ under both variants, it is **challenging to accurately compute the subpopulation allocation for samples not in the batch**. This yields the **inaccuracy when calculating the entropy term** in Eq.(2) and thus results in the performance degradation compared to SHE.
>
> | Method | CIFAR-IR100 | CIFAR-IR50 | CIFAR-IR20 |
> | :--------: | :--------: | :--------: |:--------: |
> | ERM     | 52.49%     | 55.20%     | 58.92%|
> | SHE$_\mathrm{model\ based\ V\ from\ X}$     | 53.28%     | 55.81%     | 59.72%|
> | SHE$_\mathrm{model\ based\ V\ from\ X,Y}$     | 54.08%     | 56.23%     | 60.14%|
> | SHE | 54.52%     | 56.87%     | 60.72%|
>
>
> ---
> With the aforementioned responses and revisions, we kindly ask the reviewer to check **whether our improvement has addressed the concerns of the reviewer**. We look forward to your further feedback and discussion.

---

> ### Author Response · Authors · 2023-11-19
> **Would you mind checking our responses? We welcome further discussions.**
>
> Dear Reviewer Dvju,
>
> Thank you again for your time and efforts in reviewing our submission! As the Discussion Stage and the window for paper revision are scheduled to conclude on 22nd November, we would like to offer a brief summary of our responses and updates:
>
> - Revision of the paper to improve clarity, addressing grammar errors, and refining terminologies.
> -  Clarification of the role of model complexity and finite sample considerations, supported by existing and additional empirical evidence.
> -  Inclusion of in-distribution performance, balanced-case performance, and linear probing performance to support Proposition 3.2.
> -  Providing evaluation results in scenarios where both classes and subpopulations are imbalanced.
> -  Extension of SHE for arbitrary target distributions with empirical results.
> -  Further clarification of the design of the model-based variant with additional results.
>
> **We kindly request you to check our responses with our revised version, and confirm whether they have addressed your concerns. More discussions are always welcome. Please let us know if there are any further questions or suggestions.**
>
> Sincerely,
>
> The authors of Submission 5154

---

> > ### Comment · Reviewer_Dvju · 2023-11-20
> >
> > Thank you for the revisions, clarifications, and additional experiments. I really appreciate the potential to detail and quality of the technical content. I still have some concerns with grammar and choices of terminology, which I will provide more detail for below. For now, I will change my score to 5. The main reason that I am continuing to harp on the issues with the writing is because it does impede the ability to communicate the scientific content, but I ultimately leave it to the area chair as to how much to weigh these issues in the final decision.
> >
> > For example, here is an attempted rewrite of the abstract to address some of these issues. New text is bolded and other text I am emphasizing is italicized.
> >
> > ---
> > Machine learning algorithms **learned from data with** ~under~ skew**ed** distributions usually suffer from poor generalization, especially when ~the~ *performance parity* acts as an important criterion. This is more challenging on ~the~ class-balanced data that has some hidden imbalanced subpopulations, since prevalent techniques mainly conduct ~the~ class-level calibration and cannot perform ~the~ subpopulation-level adjustment without *the explicit quantity*. Regarding ~the~ implicit subpopulation imbalance, we reveal that the key to alleviating the detrimental effect lies in ~an~ effective subpopulation discovery with proper rebalancing. We then propose a novel subpopulation-imbalanced learning method~,~ **called**~termed as~ Scatter and HarmonizE (SHE). Our method is built upon the guiding principle of optimal data partition, which involves assigning data to subpopulations in a manner that maximizes the predictive information from inputs to labels. With theoretical guarantees and empirical evidences, SHE succeeds in identifying the hidden subpopulations and encourages subpopulation-balanced predictions. Extensive experiments on various benchmark datasets show the effectiveness of SHE compared with a broad range of baselines.
> > ---
> >
> > In the abstract, there are many instances of "the" that are technically grammatically incorrect, but overall do not affect the content. Content-wise: "Machine learning algorithms under skew distributions" isn't entirely clear. A "skew distribution" isn't grammatically correct, and it would be more correct to say "skewed distribution". Furthermore, what is skewed? Is it the distribution of subpopulations or is the label distribution, or something else? It isn't clear based on the first sentence. The rewrite that I included also clarifies that it is the data that is used to fit the model that is skewed. Furthermore, it isn't clear what "performance parity" means. I would interpret this as referring to parity in performance over subpopulations, but there isn't really any way to infer that. I also don't know what "the explicit quantity" means.
> >
> > The writing in Section 1 still has similar issues as the abstract. I hope the example rewrite for the abstract is helpful for further revision. Similarly, some of the ambiguity affects understanding of the scientific content and some of it just reads as odd terminology. For example, it is very unusual to refer to majority and minority classes as "heads" and "tails" classes.

---

> ### Author Response · Authors · 2023-11-21
>
> Thank you for the feedback and further suggestions. We greatly appreciate the positive evaluation of our technical contributions and that our response has addressed the technical concerns.
>
> Following your suggestions, we have made additional careful revisions to the abstract and Section 1 in the updated version. Our efforts include correcting grammar errors, especially in the usage of "the", and replacing some terms with more lucid explanations, such as "performance parity", "explicit quantity", "head", and "tail". We would like to kindly point out that "head" and "tail" are not our subjective words, which are widely used in the area of imbalanced learning [1,2,3]. But we appreciate that the reviewer points out their limitation for understanding in a broader scope. Except for some minor grammar adjustments, all other revisions have been highlighted in blue.
>
> We kindly ask the reviewer to check our new revisions. We sincerely appreciate your diligent efforts to enhance the quality of our submission.
>
> [1] Deep long-tailed learning: A survey. TPAMI. 2023.
>
> [2] Decoupling representation and classifier for long-tailed recognition. ICLR. 2020.
>
> [3] Long-tailed classification by keeping the good and removing the bad momentum causal effect. NeurIPS. 2020.

---

> > ### Comment · Reviewer_Dvju · 2023-11-22
> >
> > Thank you for making the revisions. I will adjust my overall score to a 6.

---

> > > ### Author Response · Authors · 2023-11-22
> > >
> > > Dear Reviewer Dvju,
> > >
> > > Thank you for your time in reviewing our submission and contributing to its improvement. We sincerely appreciate your active engagement in the discussion, as well as the detailed suggestions.
> > >
> > > Best regards,
> > >
> > > The authors of Submission 5154

---

### Official Review · Reviewer_FihB · 2023-10-31

**Soundness:** 3 good
**Presentation:** 2 fair
**Contribution:** 3 good
**Rating:** 6
**Confidence:** 3

**Summary:**

The paper studied the subpopulation imbalance problem in machine learning. The setting can be described as follows: let the dataset being consist of groups $S$, and under uniform sampling, the probability $\Pr(S=s)$ varies for different realizations of $s$. If the functions $f_s: \mathcal{X}\rightarrow \mathcal{Y}$ are quite different across different groups, the machine learning algorithm that assumes a uniform $f$ would overlook the groups that contain a small number of samples.

To overcome the issue, the paper proposed a method to incorporate the subpopulation annotation in the loss function. In particular, the paper proposed the information-theoretic objective to maximize the term of $I(X;Y| v(X,Y))-I(X;Y)$, where $v(X, Y)$ is the random variable for the subpopulation annotation. To learn the optimal partition function that minimizes the term, the paper used the ‘empirical’ version of the entropy terms as the loss function, and proved that the term convergence with an additive error of $O(1/\sqrt{N})$ to the actual information ‘’gain’’ $I(X;Y| v(X,Y))$. Therefore, if we minimize the objective, we are essentially maximizing $I(X;Y| v(X,Y))-I(X;Y)$ given that the joint distribution of $(X,Y)$ is fixed.

The paper then conducted experiments on several real-world datasets to compare the proposed method with the benchmark algorithms. Experimental results show that their proposed method could outperform the benchmark algorithms in a majority of the settings, albeit the improvements are marginal.

I have a mixed feelings about this paper. On one hand, it studied a well-motivated problem, and proposed an objective function with some solid theoretical guarantees. On the one hand, the exposition of the objective and the main theorem has some non-trivial problems, and the experiment performance only offers slight improvements over the baselines. As such, I would recommend a ‘’weak accept’’ due to the pros and cons.

**Strengths:**

I think the paper studied a well-motivated problem in machine learning. Subpopulation imbalance can be viewed as an extension of the class-imbalance problem in machine learning, and the problem is harder since the ‘imbalance’ is not easily visible from the dataset. Using a training-based method is a natural idea, and essentially, the core of the paper is to train an annotation algorithm $v(X,Y)$. The paper also contains the novelty in the design of the loss function, which resorts to the tools in information theory.

The design of the objective function is justified by a theoretical analysis. Although the proofs are mostly standard applications of information theory and concentration inequalities, I appreciate the fact that they are properly written. Due to a hectic review timeline, I did not get time to verify all the calculations in Appendix B. I believe the proofs are correct with a high-level read-through.

The experiments are conducted on various datasets, and a bulk of benchmark algorithms are used in the comparison. I do appreciate the report on the error range, which justifies that the improvement is not due to statistical fluctuations.

**Weaknesses:**

A main criticism I have for this paper is that many assumptions are not explicitly stated, and the quantifiers are not properly stated in the settings and theorems. Furthermore, the paper provided no intuition on how the analysis is conducted. These problems gave me a hard time parsing the result. For instance, my first impression was that mutual information is *not* the correct notion that should be used to measure the gain for ‘subpopulation harmonization’. In particular, if $f$ is a *deterministic* function of $x$ or even a randomized function whose randomness is *independent of* $x$, then $I(X;Y)$ and $I(X;Y| v(X,Y))$ are essentially the same. (Btw $v(X,Y)$ is a random variable, which I believe you never mentioned.) I think the key in your model is that $y$ is a randomized function of $x$, and the randomness is *dependent* on the choice of $s$. However, such a fact is only clear after carefully reading the proof (!), and the whole thing reads quite confusing at first.

Additional (hidden) assumptions for the theorem in this paper include the fixed distribution of $(X,Y)$ and a fixed number of groups of subpopulations. Apart from being sloppy and not stated properly, the assumption of a fixed number of groups of subpopulations also implies we should have considerable knowledge of the dataset.

A concern about the experiment is that the improvements compared to the baseline, especially w.r.t. the very basic ERM, are too marginal. I understand that getting the SOTA performance in experiment-based machine learning is an interesting problem, however slight the improvement is. But for this specific problem, the small margin of improvement (which itself is in a low accuracy range) might have low impacts on practice.

Minor:
- In Table 1, the meaning of $p(\cdot)$ is overloaded – the ‘’class-imbalance’’ distribution is supported on the labels, while the ‘’subpopulation-imblance’’ distribution is supported on the groups of the subpopulations. I’d suggest using a different notation.

- In theorem 3.3, the Rademacher complexity of $G$ is not properly defined. Instead of simply pointing at the literature, I think the notion should be defined in the appendix with the proper quantifiers.

- Inequality (16) uses McDiarmid’s inequality, but this technical tool was never introduced. :(

**Questions:**

Most of the questions are in the ''weakness'' section. A less technical question is as follows. For the toy example in Figure 2, the original (overall) dataset is quite balanced, and an extremely skewed sampling process obtains the imbalanced training data. I understand this is important to test the performances for classification under imbalanced settings. However, can you give practical motivations to consider such a setting where we could have obtained a balanced dataset but, for some reason, have to use a very imbalanced subsampling process for the training dataset?

**Details Of Ethics Concerns:**

I do not see any necessity for ethics reviews.

---

> ### Author Response · Authors · 2023-11-16
> **Response to Reviewer FihB [1/2]**
>
> > W1: A main criticism I have for this paper is that many assumptions are not explicitly stated, and the quantifiers are not properly stated in the settings and theorems. Furthermore, the paper provided no intuition on how the analysis is conducted. These problems gave me a hard time parsing the result. For instance, my first impression was that mutual information is not the correct notion that should be used to measure the gain for ‘subpopulation harmonization’. In particular, if $f$ is a deterministic function of $x$ or even a randomized function whose randomness is independent of $x$, then $I(X;Y)$ and $I(X;Y|\nu(X,Y))$ are essentially the same. (Btw $\nu(X,Y)$ is a random variable, which I believe you never mentioned.) I think the key in your model is that $y$ is a randomized function of $x$, and the randomness is dependent on the choice of $x$. However, such a fact is only clear after carefully reading the proof (!), and the whole thing reads quite confusing at first.
>
> **A:** Thank you for your carefully reading and useful suggestions. In the revised version, we have provided explicit clarifications for certain assumptions, summarized as follows:
>
> - (In the problem formulation of Section 3.1) We explicitly clarify that the prerequisite for subpopulation imbalance is the dataset having underlying heterogeneous structures with inconsistant predictive mechanisms. That is, data distribution $p(\boldsymbol{x},y|s)$ differs across subpopulations, and $p(\boldsymbol{x}|y,s)$ may vary among certain subpopulations. Our method aims to explore and leverage subpopulations with distinct predictive mechanisms to enhance the overall prediction while safeguarding the performance of minority subpopulations.
>
> - (In Definition 3.1) We specify that $\nu(\boldsymbol{x}, y)$ is a random variable with values in $\mathcal{S}$ and explicitly state the hidden assuption of the fixed distribution and a fixed number of subpopulations.
>
> - (In the beginning part of Section 3.3) We explicitly clarify our intuitive idea: for data with implicit heterogeneous structures, we aim to partition the data so that each partition has a consistent predictive mechanism. Such an approach enhances the ability to learn more effectively from data with mixed prediction mechanisms and protects vulnerable subpopulations.
>
>
> > W2: Additional (hidden) assumptions for the theorem in this paper include the fixed distribution of $(X,Y)$ and a fixed number of groups of subpopulations. Apart from being sloppy and not stated properly, the assumption of a fixed number of groups of subpopulations also implies we should have considerable knowledge of the dataset.
>
> **A:** Thanks for the comments. We have  explicitly stated the fixed distribution and a fixed number of subpopulations in the theory part of our revised version.
>
> For the fixed number of subpopulations, our method indeed requires an empirical $K$. However, we would like to underscore two key points even if we don't have considerable knowledge of the dataset:
>
> - **Robust Performance Across $K$**: Substantial experimental evidences (**Fig. 4(a), Fig. 8(a-d)**) attest that even in the absence of knowledge about the actual number of subpopulations, our algorithm consistently achieves robust and stable improvements under various predefined values of $K$.
>
> - **Relatively Common Constraint**: Many successful clustering[1,2] and representation learning methods[3,4] have also assumed such a latent cluster number, which is mainly to provide a quantative constraint in the feature learning of DNNs. In SHE, actually, it plays a regularization role as we also have the label supervision to guide the training.
>
> [1] Yuki Markus Asano, Christian Rupprecht, and Andrea Vedaldi. Self-labelling via simultaneous clustering and representation learning. ICLR. 2020.
>
> [2] Mathilde Caron, Ishan Misra, Julien Mairal, Priya Goyal, Piotr Bojanowski, and Armand Joulin. Unsupervised learning of visual features by contrasting cluster assignments. NeurIPS. 2020.
>
> [3] Junnan Li, Pan Zhou, Caiming Xiong, and Steven C.H. Hoi. Prototypical contrastive learning of unsupervised representations. ICLR, 2021.
>
> [4] Mathilde Caron, Ishan Misra, Julien Mairal, Priya Goyal, Piotr Bojanowski, and Armand Joulin. Unsupervised learning of visual features by contrasting cluster assignments. NeurIPS. 2020.

---

> ### Author Response · Authors · 2023-11-16
> **Response to Reviewer FihB [2/2]**
>
> > W3: A concern about the experiment is that the improvements compared to the baseline, especially w.r.t. the very basic ERM, are too marginal. I understand that getting the SOTA performance in experiment-based machine learning is an interesting problem, however slight the improvement is. But for this specific problem, the small margin of improvement (which itself is in a low accuracy range) might have low impacts on practice.
>
> **A:** We would like to emphasize that our method **consistently demonstrates improvements across various datasets.** Our empirical results reveal that SHE exhibits **an average improvement of 1.5% across all five datasets** presented in Table 2 when compared to the **top-performing baseline approach** on each individual dataset. And we report error ranges to ensure statistical reliability.
>
> Additionally, SHE demonstrates **significant improvements of 4.6% in the few-split accuracy** (Table 3) and **average 3.5% in the worst-case accuracy** (Table 7), providing a more insightful evaluation of the algorithm's performance on underrepresented subgroups in subpopulation imbalance scenarios.
>
> We have summarized the mentioned experimental results in the three tables below.
>
> [**Performance on COCO, CIFAR-100, and tieredImageNet.**]
> | Method| COCO | CIFAR-IR100 | CIFAR-IR50 | CIFAR-IR20 |tieredImageNet|
> | :--------: |:--------: |:--------: |:--------: |:--------: |:--------: |
> | ERM     | 62.52±0.32% |52.49±0.27%| 55.20±0.41% |58.92±0.62% |48.23±0.27%|
> |Top-performing baseline|62.83±0.42%|53.02±0.26%|55.62±0.30%|59.19±0.37%|48.72±0.45%|
> |SHE|64.56±0.24% |54.52±0.35% |56.87±0.17% |60.72±0.41% |50.14±0.18%|
>
>
> [**Few-split Accuracy on COCO.**]
> | Method | ERM | Top-performing baseline |SHE|
> | :--------: |:--------: |:--------: |:--------: |
> | Few-split Acc     | 37.10±0.42%     | 37.67±0.24%|42.09±0.28%     |
>
> [**Worst-case performance on CIFAR.**]
>
> | Method | CIFAR-IR100 | CIFAR-IR50 | CIFAR-IR20 |
> | :--------: |:--------: |:--------: |:--------: |
> | ERM     |22.34%|26.47%|37.10%|
> |Top-performing baseline|23.83%|27.58%|38.14%|
> |SHE|27.48%|31.24%|41.19%|
>
>
> (Minor)
> > W4: In Table 1, the meaning of $p(\cdot)$ is overloaded – the ‘’class-imbalance’’ distribution is supported on the labels, while the ‘’subpopulation-imblance’’ distribution is supported on the groups of the subpopulations. I’d suggest using a different notation.
>
> > W5: In theorem 3.3, the Rademacher complexity of $G$ is not properly defined. Instead of simply pointing at the literature, I think the notion should be defined in the appendix with the proper quantifiers.
>
> > W6: Inequality (16) uses McDiarmid’s inequality, but this technical tool was never introduced. :(
>
> **A:** We do appreciate the reviewer's advice on notations and the inequlaity tool. Correspondingly, we have differentiated the notations in Table 1 by adding the proper subscripts, added the the formal definition of the Rademacher complexity of $G$ in **Appendix B.3**, and formally introduced the McDiarmid’s inequality in **Appendix B.4** in the revised version.
>
> > Q1: For the toy example in Figure 2, the original (overall) dataset is quite balanced, and an extremely skewed sampling process obtains the imbalanced training data. I understand this is important to test the performances for classiﬁcation under imbalanced settings. However, can you give practical motivations to consider such a setting where we could have obtained a balanced dataset but, for some reason, have to use a very imbalanced subsampling process for the training dataset?
>
> **A:** We would like to kindly clarify the implication of the toy example in Figure 2. It is not that we have a subpopulation-balanced dataset and sample the dataset as a subpopulation-imbalanced counterpart. It is that we usually have a subpopulation-imbalanced dataset in practice. However, **to evaluate the algorithm robustness w.r.t. latent subpopulations**, we should construct a subpopulation-balanced test set. The training set in Figure 2 simulates the real-world training scenario, and the test set reflects our desire for the algorithm to achieve good subpopulation-balanced performance. We have revised this part with a footnote to highlight this implication for clarity.

---

> ### Author Response · Authors · 2023-11-21
>
> Dear Reviewer FihB,
>
> We sincerely appreciate the effort and time you have devoted to providing constructive reviews, as well as your positive evaluation of our submission. As the deadline for discussion and paper revision is approaching,  we would like to offer a brief summary of our responses and updates:
>
> - Explicit clarification of certain assumptions.
> - Discussion and interpretation of the predefined K and numerical results.
> - Adjustment of symbols in Table 1 and a formal introduction to the mathematical tools employed.
> - Discussion on the setting of Figure 2.
>
> **Would you mind checking our responses and confirming if you have any additional questions? We welcome any further comments and discussions!**
>
> Best Regards,
>
> The authors of Submission 5154

---

> > ### Comment · Reviewer_FihB · 2023-11-22
> >
> > I have read the responses and looked at the new version. I currently do not have additional questions. I'll check with other reviews and the PCs during the discussion period to decide further actions on my review.

---

> > > ### Author Response · Authors · 2023-11-23
> > >
> > > Dear Reviewer FihB,
> > >
> > > Thank you for reviewing our responses and the revised version. We appreciate your time in providing thoughtful review comments and contributing to the improvement of this submission.
> > >
> > > Best Regards,
> > >
> > > The authors of Submission 5154

---

### Official Review · Reviewer_TvvJ · 2023-11-01

**Soundness:** 3 good
**Presentation:** 3 good
**Contribution:** 3 good
**Rating:** 8
**Confidence:** 3

**Summary:**

This work addresses the subpopulation imbalance problem where the training data consists of multiple subpopulations and their proportion is imbalanced. A novel approach, referred to as scatter and harmonize, to identifying the subpopulations and minimizing risk for each subpopulation is proposed. The authors provide a theoretical analysis of the proposed approach and demonstrate its utility through extensive experiments.

**Strengths:**

- Considering a practically important problem with clear motivation.
- Well-written and easy to follow.
- Supported by extensive experimental results.

**Weaknesses:**

- I find that this study is relevant to various problems/applications. It could be informative if the authors can clarify the relationship to the existing work, including domain generalization, algorithmic fairness, or such.
- Some details could be improved for completeness: e.g., the subpopulation imbalance problem makes sense only if $p(\boldsymbol{x},y|s)$ differs by subpopulation $s$.

**Questions:**

- I might have missed the detail, but I am not sure if the (optimal) data partition approach is completely novel. Could the authors clarify it? If not, could the authors kindly provide what approaches have been proposed? (maybe for some other problems)
- Even if the authors focused on the subpopulation imbalance problem, domain generalization and subpopulation shift problems seem highly relevant to this work. Could the authors kindly clarify the relationship between the existing methodologies to tackle the domain generalization and subpopulation shift and the subpopulation imbalance problem? Also, this work is somewhat relevant to algorithmic fairness as well. It might be helpful for future readers to relate the problems conveniently.
- For Thm 3.3, could the authors elaborate on why minimizing $\hat{\mathcal{R}}$ results in maximizing $I(X;Y,\nu(X,Y))$? It seems like it is true asymptotically but not sure with finite samples.
- Is it possible to establish an inequality between the (empirical) risk of SHE and that of ERM under the subpopulation-balanced distribution?
- This work seems to be similar to [Lahoti et al. (2020)](https://proceedings.neurips.cc/paper/2020/hash/07fc15c9d169ee48573edd749d25945d-Abstract.html), which ensures fairness with respect to maximal heterogeneity. Might be interesting to investigate the differences and similarities.
- Could the authors kindly explain why no method other than the proposed one outperforms the ERM across all datasets?

---

> ### Author Response · Authors · 2023-11-16
> **Response to Reviewer TvvJ [1/3]**
>
> > W1: I find that this study is relevant to various problems/applications. It could be informative if the authors can clarify the relationship to the existing work, including domain generalization, algorithmic fairness, or such.
>
> > Q2: Even if the authors focused on the subpopulation imbalance problem, domain generalization and subpopulation shift problems seem highly relevant to this work. Could the authors kindly clarify the relationship between the existing methodologies to tackle the domain generalization and subpopulation shift and the subpopulation imbalance problem? Also, this work is somewhat relevant to algorithmic fairness as well. It might be helpful for future readers to relate the problems conveniently.
>
> **A:** Thank you for the suggestions. Following the advice, we add more comparison with literatures on domain generalization and algorithmic fairness, and discussed the relations with subpopulation shift in Appendix A.2 and A.3. The content is summarized as below:
>
> - **Domain Generalization**: The objective of domain generalization is to extract knowledge that is invariant across diverse source domains and generalize it to novel, unseen target domains. A multitude of methods have emerged for domain generalization, broadly categorized into five groups: domain alignment[1,2], meta learning[3,4], domain hallucination[5,6], architecture-based methods[7,8], and regularization-based methods[9,10].
> - (Distinctions)
>     - In domain generalization, **domain labels are accessible** during training, whereas subpopulation annotations are not visible.
>     - The goal of domain generalization is to exhibit strong generalization performance on **unseen domains**, while the problem of subpopulation imbalance aims for a comprehensive performance **across all encountered subpopulations**.
>     - The concepts of domain and subpopulation differ in that domains are more akin to **image styles unrelated to semantics**, while subpopulations represent a kind of **semantic abstraction distinct from the class dimension**.
>     - Domain generalization methods typically aim to learn **domain-invariant features**. In contrast, our method **separates the learning of subpopulations** with different prediction mechanisms and then balancing them.
> - **Algorithmic Fairness**: When dealing with biased data, algorithms tend to make decisions based on attributes that is sensitive or should be protected(e.g., race and gender), raising concerns about fairness[11]. Various approaches[12,13,14] in algorithmic fairness aim to mitigate this issue by introducing fair constraints during the training procedure, such as demographic parity, equalized odds, etc. Some work introduces independence constraints to the objective to ensure that decisions do not rely on sensitive attributes[15,16].
> - (Distinctions)
>     - In the algorithmic fairness problem, the protected attribute annotations are **sometimes visible and sometimes not**.
>     - The key difference is that algorithmic fairness often aims to **protect specific sensitive attributes**, preventing the model from using them in decision-making. In our case, different subpopulations have different decision mechanisms, and we want to **preserve features of minority subpopulations**. In essence, while algorithmic fairness tends to **learn fewer features (excluding protected attributes)**, we aim to **learn more features (safeguarding features of the disadvantaged subpopulations)**.
>     - Additionally, the **evaluation metrics** differ between the algorithmic fairness and our problem.
> - **Subpopulation shift** can be categorized into four branches, namely **spurious correlation**, **subpopulation imbalance**, **class imbalance**, and **subpopulation generalization**[17]. Therefore, the problem in this paper falls within one of the branches of subpopulation shift.  The first three types of subpopulation shift have already been extensively discussed in Section 2 and Table 1. Subpopulation generalization closely aligns with domain generalization, with the distinction that the concept of domain is replaced by that of subpopulation.
>
>
> If you have any other recommendations or suggestions about literatures, we will appreciate and include them for further discussion.

---

> ### Author Response · Authors · 2023-11-16
> **Response to Reviewer TvvJ [2/3]**
>
> [1] Domain generalization via entropy regularization. NeurIPS. 2020.
>
> [2] Deep domain generalization via conditional invariant adversarial networks. ECCV. 2018.
>
> [3]  Feature-critic networks for heterogeneous domain generalization. ICML. 2019.
>
> [4]  Learning to learn with variational inference for cross-domain image classification. TMM. 2023.
>
> [5] Domain generalization with mixstyle. ICLR. 2021.
>
> [6] A fourier-based framework for domain generalization. CVPR. 2021.
>
> [7] Learning to balance specificity and invariance for in and out of domain generalization. ECCV. 2020.
>
> [8] Domain-specific batch normalization for unsupervised domain adaptation. CVPR. 2019.
>
> [9] Informative dropout for robust representation learning: A shape-bias perspective. ICML. 2020.
>
> [10] Domain generalization by solving jigsaw puzzles. CVPR. 2019.
>
> [11] Preventing fairness gerrymandering: Auditing and learning for subgroup fairness. ICML. 2018.
>
> [12]  Optimized pre-processing for discrimination prevention. NeurIPS. 2017.
>
> [13]  Ditto: Fair and robust federated learning through personalization. ICML. 2021.
>
> [14] Conditional learning of fair representations. ICLR. 2020.
>
> [15] Learning adversarially fair and transferable representations. ICML. 2018.
>
> [16] Learning controllable fair representations. AISTATS. 2019.
>
> [17] Change is Hard: A Closer Look at Subpopulation Shift. ICML. 2023.
>
> > W2: Some details could be improved for completeness: e.g., the subpopulation imbalance problem makes sense only if $p(\boldsymbol{x},y|s)$ differs by subpopulation $s$.
>
> **A:** Thank you for your advice, and we have refined some details to improve the completeness in the revision. Please refer to the updated submission.
>
> > Q1: I might have missed the detail, but I am not sure if the (optimal) data partition approach is completely novel. Could the authors clarify it? If not, could the authors kindly provide what approaches have been proposed? (maybe for some other problems)
>
> **A:** To our best knowledge, our optimal data partition method is completely **novel**. We kindly refer the reviewer to the [discussion](https://openreview.net/forum?id=3GurO0kRue&noteId=fCuT2n2j8P) about the **essential difference** between our method and some work involving information-theory-guided objective design. Specifically, we employ mutual information maximization **with the specific goal of uncovering heterogeneous structures within training data**. In contrast, other methods primarily focus on levaraging the information-theory-guided objective to improve representations.
>
> > Q3: For Thm 3.3, could the authors elaborate on why minimizing $\hat{\mathcal{R}}$ results in maximizing $I(X;Y;\nu(X,Y))$ ? It seems like it is true asymptotically but not sure with finite samples.
>
> **A:** Yes, there should be a more precise clarification with pointing out the asymptotic. We appreciate the thorough review by the reviewer and have adjusted the description in the revised version to avoid any confusion.
>
> > Q4: Is it possible to establish an inequality between the (empirical) risk of SHE and that of ERM under the subpopulation-balanced distribution?
>
> **A:** It is a pretty intriguing idea, and proving the relationship between SHE and ERM under subpopulation balance can provide more theoretical support for our work. However, we have to admit the current bottleneck that the optimization in SHE involves not only optimizing the model but also optimizing $\nu$, which leads to a certain level of complexity in establishing a direct connection between SHE and ERM. One possible approach is to consider that the output logit in ERM is equivalent to the sum of all heads, while SHE selects the head with the lowest loss. Under such assumption (might be strong), the first term in Eq.(1) can be lower than the risk in ERM, and the second term (the entropy term) is lower than 0. We appreciate the inspiring question of the reviewer and will conduct an in-depth exploration in the future work.
>
> For empirical evaluation, we provide the results under balanced training data below (also in **Table 18** of the revised version). Our SHE get slightly better performance even in the balanced case. According to Proposition 3.2, even though subpopulations are balanced, optimal data partition can still achieve better performance when confronted with the inherent heterogeneous structure within data.
>
>
> [**Balanced case performance**]
> | Method | CIFAR | tieredImageNet|
> | -------- | -------- | -------- |
> | ERM     | 74.32% | 68.26%|
> | SHE     | 74.75% | 68.83%|

---

> ### Author Response · Authors · 2023-11-16
> **Response to Reviewer TvvJ [3/3]**
>
> > Q5: This work seems to be similar to Lahoti et al. (2020), which ensures fairness with respect to maximal heterogeneity. Might be interesting to investigate the differences and similarities.
>
> **A:** Actually, we have conducted empirical comparisons with ARL[18] (Lahoti et al. (2020)) in the experimental section (Table 2 and 3). Please refer to a detailed discussion and comparison with ARL in Appendix A.4 of the revised version.
>
> For clarity, ARL[18] and SHE have **methodological distinctions**: ARL's core concept involves **assigning larger loss weights in areas with higher losses**, resembling the spirit of hard exmaple mining; whereas SHE aims to optimize the predictive ability, specifically the interaction information, by **partitioning data into subpopulations and concurrently rebalancing** predictions among these subpopulations.
>
> [18] Fairness without demographics through adversarially reweighted learning. NeurIPS. 2020.
>
>
> > Q6: Could the authors kindly explain why no method other than the proposed one outperforms the ERM across all datasets?
>
> **A:** The other methods except ERM mainly resort to the conventional philosophy of the imbalanced learning and supurious correlation, which cannot essentially solve the challenges of implicit subpopulation imbalance in complexity and pattern discrepency.
>
> - Class Imbalanced Learning Methods: In cases where there is no class imbalance, as seen in Table 2, there is no significant improvement over ERM. However, when both subpopulation and class imbalance coexist, as shown in Table 4 (left), there is a noticeable enhancement compared to ERM.
>
> - Spurious Correlation Methods: These methods typically assume that ERM tends to fit spurious correlations and attempt to eliminate such correlations based on ERM. In our setting, where spurious correlations are not pronounced, such an approach may impact the model's ability to learn useful features, leading to a decline in performance.
>
> Due to the intrinsic limitations of conventional imbalanced learning and the methods for supurious correlation, the inconsistent improvement (sometimes better but sometimes even worse) under the basis of ERM has been achieved, and has been surpassed by SHE through introducing the optimal data partition and the proper harmonization.

---

> ### Author Response · Authors · 2023-11-21
>
> Dear Reviewer TvvJ,
>
> We sincerely appreciate the effort and time you have devoted to providing constructive reviews, as well as your positive evaluation of our submission. As the deadline for discussion and paper revision is approaching,  we would like to offer a brief summary of our responses and updates:
>
> - Integration of discussions on literature about domain generalization, algorithmic fairnes and subpopulation shift, as well as comparison with ARL.
> - Refinement of the problem formulation and elucidation of Theorem 3.3.
> - Discussion and empirical results in balanced-case scenarios.
> - Discussion of experimental observations of baselines.
>
> **Would you mind checking our responses and confirming if you have any additional questions? We welcome any further comments and discussions!**
>
> Best Regards,
>
> The authors of Submission 5154

---

### Official Review · Reviewer_acbx · 2023-11-09

**Soundness:** 3 good
**Presentation:** 3 good
**Contribution:** 3 good
**Rating:** 6
**Confidence:** 3

**Summary:**

This paper aims to solve the subpopulation imbalance problem. The authors propose a new method named Scatter and HarmonizE (SHE), which discovers and balances the latent subpopulation in training data. Specifically, it builds on the principle of optimal data partition from information theory, which approximately uncovers the hiddle subpopulations and assigns data to subpopulations. Then, it achieves subpopulation-balanced predictions by simply applying a LogSumExp operation. Theoretical analyses are provided to support the validity of the method. Finally, experimental results illustrate the superiority of the proposed method SHE.

**Strengths:**

First of all, I have to admit that I am not an expert in the area of subpopulation imbalance, and may miss some related work.
## Originality
* To my knowledge, the data partition method for subpopulation recovery based on information theory is somewhat novel although these techniques have been widely used in other sub-fields of machine learning.
## Quality
* The proposed method is reasonable. Extensive experiments illustrate its superiority. Besides, theoretical results are also provided to support the validity of the method.
## Clarity
* Overall, this paper is well-written and the motivation is very clarified.
## Significance
* The proposed method can contribute to the community of subpopulation imbalance.

**Weaknesses:**

## Originality
* There are also many works inspired by the information theory to guide the design of training objectives in machine learning. More discussions can be added.

## Quality & Clarity
* The proposed method SHE heavily depends on the number of subpopulations $K$, which is unknown in practice.
* For the equation at the end of Section 3.1, the goal to minimize the error rate of a specific test dataset is not proper because it should be the expected error rate w.r.t. the distribution, and the one for a specific test dataset is only its unbiased estimator.

## Significance
* The proposed method may have little effect on other sub-fields of machine learning.

**Questions:**

1. In Table 1, what is the formal definition of imbalance ratio IR? I have carefully checked this paper and have not found it.

2. From the perspective of computational cost, what about the proposed method SHE against other baselines?

---

> ### Author Response · Authors · 2023-11-16
> **Response to Reviewer acbx [1/2]**
>
> > W1: There are also many works inspired by the information theory to guide the design of training objectives in machine learning. More discussions can be added.
>
> **A:** Thank you for the suggestion. In the revised version, we add a discussion and comparison of literature guiding the design of training objectives using information theory in Appendix A.1. The summary is as follows:
>
> - The **Information Bottleneck** problem [1,2] focuses on optimizing neural networks by **maximizing the relevant information about the target** while **minimizing redundant infomation**. It has gained widespread attention in recent years within the field of deep learning[3,4,5]. For instance, gradient-based methods were employed in optimizing a Deep Neural Network (DNN) to tackle the Information Bottleneck Lagrangian [6]. This approach, known as the deep variational IB (VIB), enables the system to learn stochastic representation rules, showcasing enhanced generalization capabilities and robustness to adversarial examples. A similar objective was explored in [7], where the emphasis was on promoting minimality, sufficiency, and disentanglement of representations. This disentanglement property was also harnessed for generative modeling purposes, leading to the development of the β-variational autoencoder [8].
> - **Mutual Information Maximization** (InfoMax) [9] principle is a common training objective designed to **enhance the information sharing between model outputs and target variables**. This approach is particularly popular in self-supervised learning and representation learning and have demonstrated promising empirical results [10,11]. In general, their objective is to maximize the mutual information between representations from different views of the same image. For instance, in DeepInfoMax [12], $g_1$ extracts overall features from the entire image, and $g_2$ captures local features from patches, where $g_1$ and $g_2$ are activations in different layers of the same convolution network. Contrastive Multiview Coding [13] extends the objective to incorporate multiple views, with each view corresponding to a different image modality.
> - **Comparision with our work**. The information bottleneck and mutual information maximization techniques involve the mutual information between input, representation, and label variables, aiming to **optimize the network for learning effective classifiers or generalizable representations**. In contrast to these methods, our method, distinctively, models mutual information maximization (Definition 3.1) with **the direct purpose of learning an effective data partition**, which further serves the subpopulation harmonization.
>
> If you have any other specific recommendations or suggestions for additional literature, please let us know.
>
>
> [1] The information bottleneck method. 2000.
>
> [2] The information bottleneck: Theory and applications. 2002.
>
> [3] Deep learning and the information bottleneck principle. ITW. 2015.
>
> [4] The information bottleneck problem and its applications in machine learning. IEEE J. Sel. Areas Inf. Theory. 2015.
>
> [5] How does information bottleneck help deep learning? ICML. 2023.
>
> [6] Deep variational information bottleneck. ICLR. 2017.
>
> [7] Learning optimal representations through noisy computation. IEEE Trans. Pattern Anal. Mach. Intell. 2018.
>
> [8] beta-vae: Learning basic visual concepts with a
> constrained variational framework. ICLR. 2017.
>
> [9]  Self-organization in a perceptual network. 1988.
>
> [10]  A simple framework for contrastive learning of visual representations. ICML. 2020.
>
> [11] An empirical study of training self-supervised vision
> transformers. ICCV. 2021.
>
> [12] Learning deep representations by mutual information estimation and maximization. ICLR. 2019.
>
> [13] Contrastive multiview coding. ECCV. 2020.

---

> ### Author Response · Authors · 2023-11-16
> **Response to Reviewer acbx [2/2]**
>
> > W2: The proposed method SHE heavily depends on the number of subpopulations $K$, which is unknown in practice.
>
> **A:** Although SHE requires to set an empirical $K$, we would like to underscore two key points:
>
>
> - **Robust Performance Across $K$**: Substantial experimental evidences (**Fig. 4(a), Fig. 8(a-d)**) attest that even in the absence of knowledge about the actual number of subpopulations, our algorithm consistently achieves robust and stable improvements under various predefined values of $K$.
> - **Relatively Common Constraint**: Many successful clustering[14,15] and representation learning methods[16,17] have also assumed such a latent cluster number, which is mainly to provide a quantative constraint in the feature learning of DNNs. In SHE, actually, it plays a regularization role as we also have the label supervision to guide the training.
>
> [14] Yuki Markus Asano, Christian Rupprecht, and Andrea Vedaldi. Self-labelling via simultaneous clustering and representation learning. ICLR. 2020.
>
> [15] Mathilde Caron, Ishan Misra, Julien Mairal, Priya Goyal, Piotr Bojanowski, and Armand Joulin. Unsupervised learning of visual features by contrasting cluster assignments. NeurIPS. 2020.
>
> [16] Junnan Li, Pan Zhou, Caiming Xiong, and Steven C.H. Hoi. Prototypical contrastive learning of unsupervised representations. ICLR, 2021.
>
> [17] Mathilde Caron, Ishan Misra, Julien Mairal, Priya Goyal, Piotr Bojanowski, and Armand Joulin. Unsupervised learning of visual features by contrasting cluster assignments. NeurIPS. 2020.
>
> > W3: For the equation at the end of Section 3.1, the goal to minimize the error rate of a specific test dataset is not proper because it should be the expected error rate w.r.t. the distribution, and the one for a specific test dataset is only its unbiased estimator.
>
> **A:** Thank you very much for the detailed advice. We agree that the goal should be the expected error rate, and experimental verification involves test sets corresponding to the balanced distribution for empirical performance. We have made corresponding modifications and clarifications in Section 3.1 of the revised version.
>
> > W4: The proposed method may have little effect on other sub-fields of machine learning.
>
> Indeed, our work is specifically focused on the implicit subpopulation imbalance. Nevertheless, our methodology could potentially influence robust learning under weak supervision and self supervision, as the disparity arising from the absence of annotations is inherently implicit[18,19]. Our method may provide insights for studying distribution shifts of other types as well[20]. We also aim to emphasize the significance of the community's attention to the intrinsic biases present in data, extending beyond explicit annotation dimensions.
>
> [18] SoLar: Sinkhorn Label Refinery for Imbalanced Partial-Label Learning. NeurIPS. 2022.
>
> [19] Combating Representation Learning Disparity with Geometric Harmonization. NeurIPS. 2023.
>
> [20] Change is Hard: A Closer Look at Subpopulation Shift. ICML. 2023
>
>
> > Q1: In Table 1, what is the formal definition of imbalance ratio IR? I have carefully checked this paper and have not found it.
>
> **A:** We are sorry for the confusion due to the lack of the formal definitation. Here, the imbalance ratio (IR) is defined as the ratio between the quantity of data in the subpopulation with the most samples and the subpopulation with the fewest samples in the training data, *i.e,* $\mathrm{IR} = \frac{\max\_{s\in\mathcal{S}}\sum\_{(\boldsymbol{x}\_i,y\_i,s\_i)\in\mathcal{D}}\boldsymbol{1}(s\_i=s)}{\min\_{s\in\mathcal{S}}\sum\_{(\boldsymbol{x}\_i,y\_i,s\_i)\in\mathcal{D}}\boldsymbol{1}(s\_i=s)}$. We add a formal definition and the corresponding clarifications in the revised version.
>
>
>
> > Q2: From the perspective of computational cost, what about the proposed method SHE against other baselines?
>
> **A:** Thanks for the suggestion. Considering the computational cost, SHE (according to Eq. (2) and Appendix D) incurs additional computational overhead compared to ERM due to: **1)** Weighted summation of the cross-entropy loss, **2)** Computation of empirical entropy and the regularization term, and **3)** Updating a matrix of size $BatchSize \times K$. The computational expenses for these three components are **tolerable compared with the training cost of a modern deep neural network, and smaller than most of recent methods**. The table below presents a comparison of training duration for different methods (**Table 17** of the revised version).
>
>
> [**Training time of 200 epochs on CIFAR100-IR100**.]
> | Method | Training time (Minutes) | $\Delta$ ERM |
> | :--------: | :--------: | :--------: |
> | ERM       |  89    | -  |
> |PaCo| 104| + 16.8%|
> |BCL | 110 | + 23.6%|
> |DB |  102 | + 14.6%|
> |TDE  | 95 | + 6.7%  |
> | IFL | 142  | + 60%  |
> |JTT | 101 | + 13.5%|
> |LfF | 100 | + 12.4%|
> |MaskTune| 105 | +17.9%|
> |SHE  | 98 | + 11.1% |

---

> ### Author Response · Authors · 2023-11-21
>
> Dear Reviewer acbx,
>
> We sincerely appreciate the effort and time you have devoted to providing constructive reviews, as well as your positive evaluation of our submission. As the deadline for discussion and paper revision is approaching,  we would like to offer a brief summary of our responses and updates:
>
> - Integration of discussions on literature guiding the design of training objectives using information theory.
> - Discussion and explanation of the predefined K and the significance of our work.
> - Modification of the error rate function and the formal definition of Imbalance Ratio.
> - Comparison of the training duration across different methods.
>
> **Would you mind checking our responses and confirming if you have any additional questions? We welcome any further comments and discussions!**
>
> Best Regards,
>
> The authors of Submission 5154

---

### Author Response · Authors · 2023-11-16
**General Response by Authors**

We thank reviewers for their valuable feedback, and appreciate the great efforts made by all reviewers, ACs, SACs and PCs.

For readability, we will refer to the Reviewers acbx, TvvJ, FihB, and Dvju as **R1, R2, R3, and R4**, respectively, **in the order displayed from top to bottom**.

We appreciate the positive evaluation from all reviewers, including: the problem is **well-motivated** (R1,R2,R3,R4), the method is **novel and reasonable** (R1,R2,R3), supported by **sound theoretical results** (R1,R2,R3,R4) and **extensive experiments**(R1,R2,R3,R4), **well-written and easy to follow**(R1,R2).

According to the advice, we have carefully revised our draft with the proofreading to correct some typos and mistakes. In the following, we provide a summary of our updates, and for detailed responses, please refer to the feedback of each comment/question point-by-point.

- We have incorporated a comprehensive **discussion and comparison** involving some related domains and methods, including information theory-guided objective designs (R1), domain generalization (R2), algorithmic fairness (R2), and ARL (R2). (Appendix A.1-A.4)
- We have provided **clarification and revisions** to address potential misunderstandings and confusion, including adjustments to Table 1 symbols (R3), corrections to SBER (R1), clarification of the model complexity in Figure 2 (R4), explicit statements of hidden assumptions (R2, R3), a formal introduction of the theoretical tools used (R3), an explanation of Proposition 3.2 (R4), a former definition of imbalance ratio (R1) and further interpretation of experimental results (R2, R3).
- We have **enriched our experiments** based on the existing extensive results, including computational cost (R1, Table 17),  balanced-case performance (R2, R4, Table 18), T-SNE subpopulation visualization(R4), in-distribution performance (R4, Table 19), reverse-distribution performance (R4, Table 20), and further explorations for optimizing $V$ (R4, Table 16).


The above updates in the revised version (the regular part and the appendix of totally **36 pages**) are highlighted in blue color.

We appreciate all reviewers’ time again. We are looking forward to your reply!

---

### Meta-Review · Area_Chair_kidy · 2023-12-12

**Metareview:**

The paper offers an interesting new methodology to an important problem of robustness of predictive performance on implicitly defined subpopulations with improved (albeit marginally) empirical performance in multiple experimental settings.

**Justification For Why Not Higher Score:**

The empirical improvement of the proposed methodology is a bit too incremental compared to prior methods

**Justification For Why Not Lower Score:**

The paper still offers a new methodological insight.

---

### Decision · Program_Chairs · 2024-01-16

Accept (poster)